# Safety Generalization Under Distribution Shift in Safe Reinforcement Learning: A Diabetes Testbed

**Minjae Kwon** [1]  **Josephine Lamp** [2]  **Lu Feng** [1]

## Abstract

Safe Reinforcement Learning (RL) algorithms are typically evaluated under fixed training conditions. We investigate whether training-time safety guarantees transfer to deployment under distribution shift, using diabetes management as a safety-critical testbed. We benchmark safe RL algorithms on a unified clinical simulator and reveal a *safety generalization gap*: policies satisfying constraints during training frequently violate safety requirements on unseen patients. We demonstrate that test-time shielding, which filters unsafe actions using learned dynamics models, effectively restores safety across algorithms and patient populations. Across eight safe RL algorithms, three diabetes types, and three age groups, shielding achieves Time-in-Range gains of 13–14% for strong baselines such as PPO-Lag and CPO while reducing clinical risk index and glucose variability. Our simulator and benchmark provide a platform for studying safety under distribution shift in safety-critical control domains. Code is available at https://github.com/safe-autonomy-lab/GlucoSim and https://github.com/safe-autonomy-lab/GlucoAlg.

## 1. Introduction

Reinforcement learning (RL) has achieved strong performance in complex decision-making tasks, but its deployment in safety-critical domains remains limited by the challenge of ensuring reliable safety under deployment conditions (García & Fernández, 2015; Amodei et al., 2016; Gu et al., 2024b). In response, Safe RL methods formulate control problems as Constrained Markov Decision Pro-

cesses (CMDPs) (Altman, 1999) and enforce explicit cost constraints during training using Lagrangian updates, trust-region methods, or projection-based optimization (Achiam et al., 2017; Ray et al., 2019; Yang et al., 2022). These approaches can satisfy safety constraints on the training environment, but their guarantees are distribution-dependent.

In real-world systems, deployment conditions rarely match the training environment. Dynamics vary due to changes in physical parameters, operating conditions, or population characteristics, creating distribution shift between training and test environments (Kirk et al., 2023). In such settings, policies optimized to satisfy constraints in expectation during training may violate safety requirements at deployment. Understanding whether training-time safety guarantees generalize under distribution shift remains an open and practically important question.

In this context, diabetes management provides a concrete and safety-critical testbed for studying this problem, as glucose-insulin dynamics vary substantially across individuals due to differing metabolic rates and insulin sensitivities (Dalla Man et al., 2007; Battelino et al., 2019). While prior RL approaches have explored diabetes control in simulation and offline settings (Fox et al., 2020; Zhu et al., 2023), they typically assume matched training and testing dynamics. Consequently, the robustness of safety constraints under physiological variability remains largely unexplored. Furthermore, existing studies often prioritize standard reward-based objectives over explicit safety constraints and focus mainly on Type 1 diabetes, leaving safety-constrained adaptation across diverse clinical populations unaddressed.

In this work, we show that this assumption is consequential. Across eight commonly used safe RL algorithms, we observe a *safety generalization gap*: policies that achieve high Time-in-Range and low clinical risk on the training patients frequently degrade and violate safety margins when evaluated on unseen patients. This gap persists across diabetes types, age groups, and algorithm families, indicating a structural limitation of training-time constraint enforcement.

Our goal is not to propose a new Safe RL algorithm, but to provide a rigorous framework for studying and mitigating safety failures under distribution shift. We make three contributions:

The views expressed in this paper are solely those of the authors and do not reflect the official policy or position of Dexcom, Inc. [1]Department of Computer Science, University of Virginia [2]Dexcom. Correspondence to: Lu Feng <lu.feng@virginia.edu>.

*Proceedings of the 43rd International Conference on Machine Learning*, Seoul, South Korea. PMLR 306, 2026. Copyright 2026 by the author(s).

1. **Unified Clinical Simulator.** We introduce a simulator that supports Type 1 and Type 2 diabetes with pump and non-pump therapies, modeling therapeutic *decision support* rather than direct control. It captures realistic sources of deployment mismatch, including latent patient variability and partial adherence, enabling controlled evaluation of safety generalization.

2. **OOD Safety Benchmark.** We create a benchmark for evaluating safe RL algorithms under physiological distribution shift. By testing eight safe RL algorithms, we isolate the gap between training-time constraint satisfaction and test-time safety. This provides a reusable testbed for evaluating out-of-distribution safety generalization in safety-critical RL.

3. **Test-Time Predictive Shielding.** We study a runtime shielding mechanism that acts as an algorithm-agnostic safety wrapper to mitigate safety generalization issues under distribution shift. To support accurate forecasting under patient variability, we introduce Basis-Adaptive Neural ODEs (BA-NODE), a continuous-time predictor combining multivariate history encoding (Liu et al., 2024), neural ODE (Chen et al., 2018), and function-space conditioning (Ingebrand et al., 2024).

Across 72 experimental settings, predictive shielding consistently recovers safety margins and improves clinical outcomes, showing Time-in-Range gains of up to 14% for strong baselines while reducing risk and glucose variability. These results demonstrate that test-time verification provides a practical, algorithm-agnostic complement to Safe RL for bridging the safety generalization gap.

**Navigation.** This paper is organized as follows. We first review relevant background and related work. Section 3 introduces a unified diabetes simulation environment designed to evaluate safety generalization under physiological distribution shift. Because predictive shielding requires forecasting future safety risks, Section 4 presents a personalized blood glucose dynamics prediction model used by the shield. Section 5 describes the design of the predictive shielding mechanism and how it mitigates safety generalization failures at test time. Section 6 reports benchmark results on OOD safety generalization, including evaluations of both the shielding mechanism and the dynamics predictor. Section 7 concludes the paper.

## 2. Background & Related Work

**Safe Reinforcement Learning under Fixed Dynamics.** Safe Reinforcement Learning is commonly formalized using the Constrained Markov Decision Process (CMDP) framework (Altman, 1999), in which an agent maximizes expected return while satisfying cumulative safety constraints.

Given a CMDP $(\mathcal{S}, \mathcal{A}, P, R, C, \gamma, d)$, representing the state space, action space, transition dynamics, reward function, cost function, discount factor, and cost limit, respectively. The objective is to learn a policy $\pi$ that maximizes $\mathbb{E}_\pi[\sum_t \gamma^t R(s_t, a_t)]$ subject to $\mathbb{E}_\pi[\sum_t \gamma^t C(s_t, a_t)] \leq d$. Classic approaches enforce this constraint during training using Lagrangian relaxation (Ray et al., 2019; Tessler et al., 2019) or trust-region methods such as Constrained Policy Optimization (CPO) (Achiam et al., 2017). Subsequent work has proposed projection-based updates, recovery policies, first-order constrained optimization, sample-manipulation strategies, and model-based approach (Zhang et al., 2020; Yang et al., 2020; Xu et al., 2021; Yang et al., 2022; Gu et al., 2024a; As et al., 2025). These methods are designed to achieve strong performance and safety satisfaction under a fixed environment. However, they do not address robustness to changes in the underlying transition dynamics at deployment.

**Generalization and Distribution Shift.** Real-world deployment requires robustness to distribution shift, yet RL agents often overfit to training dynamics, degrading out-of-distribution (OOD) performance, which is mainly measured by reward-based metrics (Cobbe et al., 2019; 2020; Song et al., 2020). While methods like domain randomization (Tobin et al., 2017) and adversarial training (Pinto et al., 2017) can improve robustness under distribution shift, these studies mainly focus on reward, leaving the maintenance of *safety constraints* under distribution shift largely unexplored (Packer et al., 2019; Kirk et al., 2023). Safe meta RL (Khattar et al., 2023; Guan et al., 2024; Xu & Zhu, 2025) adapt to new tasks, updating parameters to satisfy safety constraints. However, such adaptation is often interaction-heavy and may violate safety constraints during online learning. This gap is particularly highlighted in medical control, which presents a harder challenge than standard robotics benchmarks: shifts are *latent* and structural (e.g., unobservable metabolic traits) rather than observable parameters (e.g., size in robots), and ethical constraints prohibit trial-and-error retraining (Dulac-Arnold et al., 2021). Consequently, medical control provides a clear testbed for studying *safety generalization*, where safety must be enforced at test time rather than guaranteed during training.

**Runtime Safety Enforcement and Shielding.** Complementary to training-time constraints, *shielding* provides a test-time safety mechanism that filters unsafe actions without requiring policy retraining (Alshiekh et al., 2018; Carr et al., 2023). Shielding approaches typically relied on formal logic or access to known system dynamics to verify safe action sets (Alshiekh et al., 2018; Yang et al., 2023), which limits their applicability in settings with uncertain or partially observed dynamics.

Recent work has relaxed this assumption by incorporating predictive models into the safety check. In particular, methods based on adaptive conformal prediction enable uncertainty-aware shielding under perception noise and modeling errors (Sheng et al., 2024b;a; Scarbro et al., 2025). While these approaches reduce safety violations at runtime under fixed dynamics, their formal guarantees often do not extend to distributional shift in the underlying environment.

Our approach follows this predictive paradigm but targets a distinct challenge: safety generalization under distribution shifts. Unlike robotic settings where uncertainty often arises from observable parametric variation, medical control involves unobserved, patient-specific metabolic dynamics. We proposes a Basis-Adaptive Neural ODE to predict glucose trajectories through patients-specific adaptation. Combining with this, our proposed shielding improves safety in clinical metrics without retraining the policy under physiological distribution shift.

## 3. Unified Diabetes Simulator

We develop a unified clinical simulator for *therapeutic decision support* in Type 1 and Type 2 diabetes, supporting both pump and non-pump therapies. Unlike standard benchmarks (Man et al., 2014; Zhu et al., 2023; Marchetti et al., 2025) that assume direct, continuous control of basal insulin, our simulator models the interaction between a recommender agent and a patient. Reflecting real-world practice in which basal rates are prescribed and adjusted infrequently (Kuritzky et al., 2019; Mehta et al., 2021), the agent proposes discrete interventions such as bolus insulin and meal intake, while execution depends on patient compliance and physiological constraints. Full physiological equations and implementation details are provided in Appendix A, including ODE system, blood glucose dynamics, and the unified framework for T1D and T2D cohorts.

### 3.1. Physiological Model

The simulator builds on the UVA–Padova family of mechanistic glucose–insulin models (Dalla Man et al., 2007; 2009) and supports three clinical settings: (i) Type 1 diabetes with pump therapy, (ii) Type 2 diabetes with pump therapy, and (iii) Type 2 diabetes without pump therapy. We implement a *hybrid T2D formulation*, combining Hovorka secretion dynamics with Dalla Man transport models to capture insulin resistance. Patient variability is driven by physiological parameters affecting sensitivity, absorption, and clearance, all of which remain unobserved by the agent.

### 3.2. MDP Interface

**Observation and Action Interface.** At each step, the agent receives a vector of clinically observable quantities derived from continuous glucose monitor (CGM) readings, insulin-on-board, and meal history. The policy outputs *recommended* bolus and meal interventions, which are filtered through a patient acceptance model that enforces safety gates. Basal insulin is fixed at patient initialization and calibrated using weight-based dosing. Details are provided in Appendix A.3 and Appendix A.4, including 14-dimensional observation space, and the logic that regulates the action interface.

**Reward and Cost Design.** We use predicted glucose trajectories to evaluate the long-term effects of actions when computing reward and cost, capturing delayed phenomena such as insulin stacking and rebound hyperglycemia that are not reflected by immediate measurements. The risk term is asymmetric, penalizing hypoglycemia more strongly than hyperglycemia due to its higher acute clinical danger (Cryer, 2008; Battelino et al., 2019).

The cost function additionally discourages overly frequent interventions through action-regularization penalties, reflecting clinical practice where excessive corrections increase patient burden (Vijan et al., 2005). We empirically verify that cumulative reward and cost align with standard clinical metrics, including Time-in-Range and Risk Index (Appendix A.6). Details are provided in Appendix A.5, covering two-hour predictive horizon delta-risk calculations, the penalties for exceeding daily caps and frequent interventions, and the early termination for critical glycemic events.

## 4. Personalized Dynamics Learning via Basis-Adaptive Neural ODEs

Effective safety shielding requires a dynamics model that captures smooth glucose–insulin evolution while remaining robust to large inter-patient variability. To this end, we propose the *Basis-Adaptive Neural ODE (BA-NODE)*, a dynamics model that combines continuous-time latent evolution with function-space conditioning.

BA-NODE represents glucose dynamics using a continuous-time latent system governed by a neural ordinary differential equation (Chen et al., 2018). Let $h(t)$ denote a latent physiological state whose evolution is parameterized by a learnable vector field $\frac{dh(t)}{dt} = f_\theta(h(t), t)$. This formulation provides a natural model for smooth physiological dynamics.

To support personalization, BA-NODE adopts function-space conditioning principle of Function Encoder (FE) (Ingebrand et al., 2024). Instead of learning a single patient-specific dynamics model, BA-NODE learns a shared set of latent dynamical components and adapts to individual patients by forming linear combinations of these components using context-dependent weights computed at inference time.

A direct comparison with original FE framework is not applicable, as FE is formulated for static regression and lacks the history-dependent representations required for glucose–insulin modeling. BA-NODE bridges this gap by combining a variate-aware history encoder with a latent ODE ensemble, transforming static function bases into a set of dynamical basis trajectories. Concretely, the architecture comprises three modules: (i) an Inverted Transformer (ITransformer) for variate-aware history encoding (Liu et al., 2024), (ii) a Latent Neural ODE ensemble for continuous evolution, and (iii) a Function-Space adaptation mechanism (Ingebrand et al., 2025).

### 4.1. Variate-Aware History Encoding

Given a window of past observations $x_{1:T} \in \mathbb{R}^{T \times d}$, we encode physiological history using an ITransformer (Liu et al., 2024). Each physiological variate (e.g., glucose, insulin, carbohydrates) is treated as a token, allowing the encoder to explicitly model cross-variate interactions.

Each variate $v$ is treated as a token and embedded into $z_v \in \mathbb{R}^{d_{model} \times 1}$. Cross-variate self-attention yields token-wise representations, which are projected to a per-variate summary vector $z \in \mathbb{R}^{d \times 1}$. A projection head $g_\psi$ maps this vector to the initial latent state: $h_0 = g_\psi(z) \in \mathbb{R}^{d_{latent} \times 1}$.

### 4.2. Latent Dynamics via an ODE Ensemble

Latent evolution is governed by an ensemble of $K$ neural vector fields $\{f_{\theta_k}\}_{k=1}^K$, each representing a candidate mode of glucose dynamics. Forward integration is performed using a fourth-order Runge–Kutta (RK4) solver. For each time step, the model computes $K$ candidate updates:
$$\Delta h_t^{(k)} = \text{RK4}(f_{\theta_k}, h_t) \in \mathbb{R}^{d_{latent} \times 1}, \quad k = 1, \dots, K.$$

Each update defines a candidate next latent state $\tilde{h}_{t+1}^{(k)} = h_t + \Delta h_t^{(k)} \in \mathbb{R}^{d_{latent} \times 1}$. These candidates are combined through a shared linear projection: $h_{t+1} = \left[\tilde{h}_{t+1}^{(1)}, \dots, \tilde{h}_{t+1}^{(K)}\right] W_{proj}$ where $W_{proj} \in \mathbb{R}^{K \times 1}$ is learned jointly with the vector fields. This ensemble formulation increases representational capacity while preserving a single coherent latent trajectory, enabling the model to capture the nonlinear dynamics.

### 4.3. Context-Conditioned Function-Space Combination

The $K$ latent rollouts induced by the ODE ensemble are interpreted as a set of *basis trajectories* $\{G_k(\cdot)\}_{k=1}^K$, where each basis produces a horizon-length glucose changes $\Delta \hat{y}_{T+1:T+P}$. We concatenate these into $G(x) = \left[G_1(x), \dots, G_K(x)\right] \in \mathbb{R}^{P \times K}$. Rather than selecting a single trajectory, BA-NODE adapts to a specific patient by forming a weighted combination of these basis trajectories. For each patient we collect a set of $N$ context windows

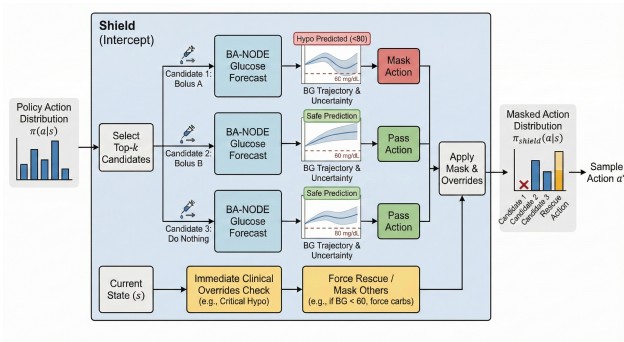

*Figure 1.* **Predictive Verification Flow**. The shield intercepts the policy's action distribution and identifies the top-$k$ candidate actions. It queries BA-NODE dynamics model to predict the blood glucose trajectory for each candidate. Actions resulting in predicted safety violations (hypo- or hyperglycemia) are masked. Additionally, immediate clinical overrides (e.g., rescue carbs) are applied alongside these predictive checks.

$\{(x_{ctx}^{(n)}, y_{ctx}^{(n)})\}_{n=1}^N$, with $x_{ctx}^{(n)} \in \mathbb{R}^{T \times d}$ and $y_{ctx}^{(n)} \in \mathbb{R}^{P \times 1}$ (delta glucose over the prediction horizon). The model maps each context to $K$ basis rollouts $G(x_{ctx}^{(n)}) \in \mathbb{R}^{P \times K}$. We then stack all $N$ contexts into a single design matrix $\tilde{G} \in \mathbb{R}^{(NP) \times K}$ by concatenating the $P$-step outputs across contexts, and likewise stack targets into $\tilde{y}_{ctx} \in \mathbb{R}^{NP \times 1}$.

Following Ingebrand et al. (2025), patient-specific mixing weights $w^\star \in \mathbb{R}^K$ are computed via regularized least squares: $w^\star = \arg\min_w \|\tilde{G}w - \tilde{y}_{ctx}\|_2^2 + \lambda \|w\|_2^2$. Future absolute glucose is predicted by applying $w^\star$ to the basis and cumulating the resulting increments from the last observed value $y_T$: $\hat{y}_{T+P} = y_T + \sum_{i=1}^P (G(x_{pred}) w^\star)_i$. Additional analyses of coefficient behavior, adaptation ablations, and training details are provided in Appendices F.6, F.7, and F.8.

## 5. Test-Time Predictive Shielding

Safe RL algorithms enforce constraints by optimizing a policy on a fixed training distribution. However, in medical deployment, agents inevitably face physiological distribution shifts, such as unseen insulin sensitivities or varying absorption rates, which can invalidate the safety boundaries learned during training. Training-time optimization alone is insufficient under unseen dynamics. We introduce a *test-time predictive shield*, a runtime wrapper that intercepts policy actions, forecasts their future trajectories, and preemptively blocks any recommendations flagged as dangerous.

### 5.1. Shielding Architecture: Action Masking

Let $\pi_\theta(a|s)$ be the pre-trained policy. The shield constructs a valid action mask $M(s, a)$ that applies finite logit bonuses and penalties based on static clinical rules and dynamic

predictive verification. The shielded policy is given by:

$$\pi_{\text{shielded}}(a|s) = \text{Softmax}(\log \pi_\theta(a|s) + M(s,a)). \quad (1)$$

The shield discourages risky actions while preserving a nonzero probability mass over alternatives.

## 5.2. Shield Decision Logic

The shield evaluates three distinct conditions using two clinically motivated thresholds. The rescue threshold $G_{\text{rescue}} = 60 \, \text{mg/dL}$ implements a model-free, clinically mandated emergency intervention, adding a safety margin above severe hypoglycemia ($54 \, \text{mg/dL}$) and remaining independent of predictive accuracy. In contrast, the predictive safety threshold $G_{\text{shield}}^\downarrow = 80 \, \text{mg/dL}$ introduces a safety margin above the clinical hypoglycemia limit ($70 \, \text{mg/dL}$) to account for model prediction errors.

1. **Critical Rescue.** If $BG_t < G_{\text{rescue}}$, the shield forces rescue carbohydrates (e.g., 15g) and blocks all insulin actions.
2. **Predictive Safety.** If $BG_t \geq G_{\text{shield}}^\downarrow$ but candidate actions forecast near-term violations, the shield applies predictive verification to penalize unsafe actions.
3. **Minimal Intervention.** If no safety risks are detected, the agent's action is executed without modification.

**Gating Mechanism.** In practice, we pause predictive verification near the safety boundary to prevent instability. When $BG_t$ is in the transition zone $[G_{\text{rescue}}, G_{\text{shield}}^\downarrow)$, the model can be sensitive to small noise, which causes the shield to over-react with repeated, unnecessary corrections (e.g., frequently recommending rescue carbs). To avoid this, we disable the shield in this region and trust the base policy's smooth control, re-engaging the shield only when $BG_t \geq G_{\text{shield}}^\downarrow$ to prevent future risks.

## 5.3. Predictive Safety Verification

Static rules often fail to prevent delayed safety risks caused by insulin stacking. To address this, the shield uses predictive verification. The shield selects the top-$k$ bolus candidates under $\pi_\theta$ and pairs each with all discrete meal levels. For each candidate action $a$, BA-NODE dynamics model predicts a blood glucose trajectory $\widehat{BG}_{t:t+H}(a)$ over a finite horizon $H$.

We distinguish between clinical failure limits, which define unsafe physiological states, and shield thresholds, which enforce conservative action pruning. Let $G_{\text{fail}}^\downarrow$ and $G_{\text{fail}}^\uparrow$ denote hypoglycemia ($< 70 \text{mg/dL}$) and hyperglycemia limits ($> 180 \text{mg/dL}$), respectively. The shield enforces stricter thresholds $G_{\text{shield}}^\downarrow > G_{\text{fail}}^\downarrow$ and $G_{\text{shield}}^\uparrow < G_{\text{fail}}^\uparrow$ to account for prediction errors. We define the extremal predicted glu-

cose values:

$$m(a) := \min_{\tau \in [t,t+H]} \widehat{BG}_\tau(a), \, M(a) := \max_{\tau \in [t,t+H]} \widehat{BG}_\tau(a).$$

The shield penalizes actions based on these thresholds:

- **Hypo prevention:** Penalize $a$ if $m(a) < G_{\text{shield}}^\downarrow$.
- **Hyper prevention:** Penalize $a$ if $M(a) > G_{\text{shield}}^\uparrow$.

## 5.4. Probabilistic Safety Bounds

We now formalize the safety guarantee induced by predictive pruning under a bounded prediction error assumption. The key idea is to interpret the shield thresholds as incorporating an explicit safety margin relative to clinical failure limits.

**Definition 5.1** (One-sided $(\varepsilon, \alpha)$-reliability). Fix a horizon $H$. A predictor is said to be $(\varepsilon, \alpha)$-reliable for hypoglycemia if, for any action $a$,

$$\Pr\left(\max_{\tau \in [t,t+H]} \left(\widehat{BG}_\tau(a) - BG_\tau(a)\right) \leq \varepsilon\right) \geq 1 - \alpha. \quad (2)$$

That is, with probability at least $1 - \alpha$, the predictor does not overestimate blood glucose by more than $\varepsilon$ over the horizon.

**Theorem 5.2** (Hypoglycemia Safety). *Suppose the shield uses a stricter pruning threshold $G_{\text{shield}}^\downarrow = G_{\text{fail}}^\downarrow + \varepsilon$. If the predictor is $(\varepsilon, \alpha)$-reliable, then any action $a$ permitted by the shield satisfies*

$$\Pr\left(\min_{\tau \in [t,t+H]} BG_\tau(a) \geq G_{\text{fail}}^\downarrow\right) \geq 1 - \alpha. \quad (3)$$

See Appendix B for the complete proof and extension to hyperglycemia safety.

# 6. Experiments

Our evaluation assesses the proposed framework on two levels: (i) the predictive accuracy of the BA-NODE dynamics predictor, and (ii) the safety improvements provided by the shield across different RL agents. To ensure rigorous comparison, we report results using clinical metrics, rather than the shaped reward and cost signals used for training.

**Evaluation Metrics.** We assess performance using standard clinical outcome metrics that are independent of the training reward and cost.

- **Time in Range (TIR).** The percentage of time blood glucose remains within the target interval $[70, 180]$ mg/dL (Battelino et al., 2019).
- **Coefficient of Variation (CV).** For a glucose trace $BG_{1:T}$, we quantify stability using the Coefficient of Variation ($CV = \sigma/\mu$), where $\mu$ and $\sigma$ are the mean and standard deviation of the sequence (Battelino et al., 2019).

- **Risk Index.** We compute the instantaneous risk $r_t = 10f(G_t)^2$ using the symmetrizing transformation $f(G) = 1.509(\ln(G)^{1.084} - 5.381)$ (Kovatchev et al., 1997). We report the mean risk index $\frac{1}{T}\sum r_t$. This metric applies asymmetric penalties to hypoglycemia and hyperglycemia, capturing acute clinical danger not reflected by TIR alone.

We verify that cumulative reward and accumulated cost correlate positively with TIR and Risk Index, respectively. See Appendix A.6 for details.

**Baselines.** We benchmark diverse safe RL algorithms within the proposed unified diabetes simulator, evaluating each policy both with and without our predictive shielding.

- **PPO-Lag**: Proximal Policy Optimization with Lagrangian updates for both reward and constraint (Schulman et al., 2017; Ray et al., 2019).
- **TRPO-Lag**: Trust Region Policy Optimization with second-order reward updates and first-order constraint updates (Schulman et al., 2015; Ray et al., 2019).
- **CPO**: Constrained Policy Optimization with joint second-order updates to enforce linearized cost constraints (Achiam et al., 2017).
- **RCPO**: Reward Constrained Policy Optimization, which uses policy gradients to optimize a reward function penalized by safety violations (Tessler et al., 2019).
- **FOCOPS**: First-Order Constrained Optimization in Policy Space, which solves a constrained policy optimization problem using first-order gradients (Zhang et al., 2020).
- **PCPO**: Projection-based Constrained Policy Optimization, which decouples reward and constraint learning via policy projection (Yang et al., 2020).
- **CRPO**: Constrained Reinforcement Learning via Projection, which alternates between reward optimization and cost minimization (Xu et al., 2021).
- **CUP**: Constrained Update Projection, which enforces safety via projected policy updates with theoretical guarantees (Yang et al., 2022).
- **Rule-Based Shield (RBS)**: A safety wrapper mimicking Low Glucose Suspend (LGS) systems, which overrides actions based on fixed glucose and Insulin-on-Board (IOB) thresholds (Bergenstal et al., 2013). See Appendix D for details.

**Training Setup.** We train distinct policies for each specific condition (T1D, T2D, T2D without Pump) and age group (Child, Adolescent, Adult), resulting in a total of 9 base policies for each safe RL algorithm. To model the challenge of real-world deployment where patient-specific data

| Algorithm | Training Patient (ID) | | Unseen Patients (OOD) | | Generalization Gap | |
|---|---|---|---|---|---|---|
| | TIR (%) ↑ | Risk Index ↓ | TIR (%) ↑ | Risk Index ↓ | Δ TIR | Δ Risk |
| CPO | 87.28 ± 2.22 | 4.09 ± 0.65 | 76.73 ± 3.98 | 5.72 ± 1.02 | -10.55 | +1.62 |
| CUP | 89.36 ± 3.12 | 3.27 ± 0.79 | 77.70 ± 4.90 | 5.88 ± 1.42 | -11.66 | +2.61 |
| FOCOPS | 88.08 ± 0.44 | 3.50 ± 0.14 | 75.42 ± 0.38 | 6.27 ± 0.20 | -12.66 | +2.77 |
| CRPO | 84.94 ± 4.26 | 4.12 ± 0.76 | 74.50 ± 2.27 | 6.14 ± 0.53 | -10.45 | +2.03 |
| PCPO | 36.74 ± 1.84 | 20.34 ± 0.33 | 38.75 ± 0.52 | 19.97 ± 0.33 | +2.01 | -0.37 |
| PPOLag | 85.29 ± 4.06 | 4.07 ± 0.95 | 75.20 ± 2.63 | 6.23 ± 0.58 | -10.09 | +2.16 |
| RCPO | 82.98 ± 6.12 | 4.70 ± 1.51 | 74.09 ± 2.77 | 6.40 ± 0.62 | -8.89 | +1.70 |
| TRPOLag | 82.79 ± 4.65 | 4.80 ± 0.88 | 74.43 ± 2.38 | 6.60 ± 0.39 | -8.37 | +1.79 |

*Table 1.* **The Safety Generalization Gap (Unshielded Baselines).** Performance comparison between the Training Patient (in-distribution, denoted by ID) and Unseen Patients (OOD).

is scarce, we restrict training to a *single* representative patient per cohort (specifically `Child#01`, `Adolescent#01`, and `Adult#01`). This prevents the agent from memorizing diverse physiological parameters during optimization and forcing it to learn generalized dynamics from limited interaction. See Appendix E for training hyperparameters, hyperparameter optimization for baselines, and a summary of training dynamics.

**Evaluation Setup.** We evaluate the policies under *zero-shot* conditions to assess the generalization of training-time safety constraints:

- **Parameter Generalization:** Agents are tested on the *full* cohort of 9 distinct patients (`Patients #02-#10`) per diabetes type, none of which were used during training. This introduces physiological distributional shifts (e.g., unseen insulin sensitivities and absorption rates). See Appendix A.2 for details, including the distribution of key physiological parameters across age and cohorts, and the age-dependent shifts in basal metabolic states.
- **Horizon Generalization:** During training, we train the model using 1 day scenarios. At evaluation time, we extend the episode length to 7 simulated days to assess long term stability and error accumulation.

We report the performance gains ($\Delta$) measured for shielding relative to the unshielded baselines

### 6.1. OOD Safety Benchmark

Before evaluating our proposed shielding mechanism, we first examine whether training-time safety guarantees provided by existing Safe RL algorithms transfer to unseen patients. We use our unified simulator to benchmark eight safe RL baselines under physiological distribution shift.

Table 1 shows a *safety generalization gap*: policies that achieve high TIR and low Risk Index on the training patients show substantial degradation when deployed on physiologically distinct patients. Despite all algorithms being trained to satisfy safety constraints (e.g., Risk Index < 5 and TIR > 80%), deployment under shift leads to consistent violations, resulting in a TIR reduction of more than 10%.

This failure validates the simulator as a rigorous OOD safety

| Model | MAE ↓ | FDE ↓ | RMSE ↓ |
|---|---|---|---|
| ITransformer | $4.11 \pm 0.10$ | $5.39 \pm 0.16$ | $7.25 \pm 0.29$ |
| NODE | $3.18 \pm 0.60$ | $4.11 \pm 1.23$ | $5.10 \pm 1.45$ |
| BA-NODE | $\mathbf{2.82 \pm 0.01}$ | $\mathbf{3.63 \pm 0.03}$ | $\mathbf{4.42 \pm 0.02}$ |

*Table 2.* **Dynamics Model Performance (24-step Horizon).** Evaluation of predictive accuracy using MAE, FDE, and RMSE. BA-NODE outperforms baselines with the lowest error.

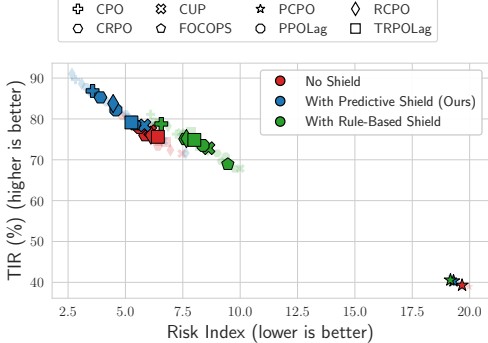

*Figure 2.* Trade-off between Risk Index (lower is better) and TIR (higher is better) across algorithms. Faint points show per-seed results; larger outlined markers show seed-averaged means.

benchmark, demonstrating that training-time constraint satisfaction does not reliably translate to test-time safety and motivating the need for deployment-time mechanisms such as predictive shielding.

**Research Questions.** Our experiments are guided by the following research questions:

- **RQ1:** Does the proposed Basis-Adaptive Neural ODE improve blood glucose prediction accuracy compared to other dynamics models?
- **RQ2:** Does predictive shielding consistently improve safety-performance across diverse safe RL algorithms?
- **RQ3:** Does predictive shielding remain effective under physiological distribution shifts with distinct dynamics?

### 6.2. RQ1: BA-NODE Improves Dynamics Prediction Accuracy

**Dataset and Prediction Task.** We train and evaluate all dynamics models on trajectories generated by the unified clinical simulator. For each cohort (adult, adolescent, pediatric), we collect 15 days of glucose trajectories generated by a pre-trained policy and collect an additional 5 days for evaluation. Models are trained to predict blood glucose from historical observations, using a fixed history window and evaluated over multi-step prediction horizons up to 120 minutes (24 timesteps).

**Analysis.** To estimate safety violations, predictive shielding benefits from accurate multi-step dynamics forecasts. We therefore first evaluate whether BA-NODE improves glucose prediction accuracy relative to baselines. Table 2 compares BA-NODE against a standard NODE and ITransformer over a 120-minute prediction horizon. We report Mean Absolute Error (MAE), Root Mean Squared Error (RMSE), and Final Displacement Error (FDE), averaged across evaluation seeds.

BA-NODE achieves the lowest error and variance across all metrics, demonstrating superior training stability and robustness compared to baselines. This accuracy directly reduces forecast drift, ensuring the shield can accurately evaluate candidate trajectories and reliably filter out dangerous action choices before they are executed. By maintaining low error over 24 timesteps prediction window, the model enables the shield to distinguish between safe and risky policies

with high confidence. See Appendix F for detailed ablations regarding prediction horizons and training dynamics.

### 6.3. RQ2: Shielding Improves Safety Across Algorithms

We first investigate whether predictive shielding offers a superior safety-performance trade-off compared to static heuristics across the entire diverse population of patients. Figure 2 visualizes the aggregated results across all diabetes types (T1D, T2D, T2D without pump) and cohorts, plotting Time-in-Range (TIR, y-axis) against Risk Index (x-axis).

**Comparison to rule-based shielding.** Across most settings, the predictive shield outperforms a static rule-based shield. Rule-based shield often *degrades* performance relative to the unshielded policy. This behavior arises from rigid heuristic interactions: aggressive bolus corrections for hyperglycemia can overshoot, inducing hypoglycemia, which then triggers compensatory meal recommendations. The resulting feedback loop produces large glucose oscillations and elevated variability. See Figure 7 in Appendix C for trajectory examples.

**Failure modes and limits.** An exception occurs when the underlying controller performs poorly, such as CUP and PCPO. For example, a controller trained by PCPO achieves TIR below 40% and a Risk Index around 20. In such cases, the rule-based shield can outperform the predictive shield. This is expected because the predictive shield operates by adjusting the action distribution proposed by the base policy. Hence, this remains partially constrained by its probability mass. When the base policy performs poorly, purely heuristic overrides may perform better.

Overall, these results demonstrate that predictive shielding provides a robust, algorithm-agnostic mechanism for improving safety and clinical performance at test time, consistently outperforming both unshielded policies and static rule-based alternatives. Detailed per-cohort and per-algorithm

metrics, including TIR, Risk Index, hypoglycemic and hyperglycemic event rates, and analyses of glucose variability and rescue dynamics, are reported in Appendix G.2.

### 6.4. RQ3: Robustness Under Distinct Dynamical Regimes

While RQ2 demonstrated global efficacy, we now examine whether predictive shielding remains effective across distinct physiological regimes. We decompose performance by diabetes type and additionally report glycemic variability using CV. Clinical evidence demonstrates that glucose volatility contributes to complications and hypoglycemia risk independently of TIR (Hirsch & Brownlee, 2005; Monnier et al., 2008), making trajectory smoothness a critical dimension of control (Battelino et al., 2019). This analysis allows us to assess whether predictive shielding improves both safety and stability.

**Type 1 Diabetes (T1D).** Table 3 reports results for the T1D cohort (no endogenous insulin; exogenous pump control). Predictive shielding shows consistent improvements across safe RL algorithms, with an average $\Delta$TIR of $+6.08\%$. Importantly, CV is reduced for all agents (e.g., FOCOPS achieves $\Delta$CV $-4.26\%$). This indicates that the shield not only prevents threshold violations but also smooths glucose trajectories, mitigating the oscillatory behavior.

**Type 2 Diabetes (T2D).** Table 4 presents results for the T2D cohort (insulin resistance with residual secretion). In this setting, the benefits of predictive shielding are more visible. Predictive Shielding shows substantial gains: PPO-Lag improves TIR by $+14.15\%$, while CPO improves by $+13.54\%$. Regarding CV, CPO has the most gain achieving $-6.68\%$ drops followed by PPO-Lag's $-5.5\%$ drops in CV.

**T2D without Pump.** Table 5 evaluates the T2D cohort without pump, where patients rely on residual endogenous secretion and bolus-only corrections. In this setting, gains are less consistent across algorithms. For example, PPO-Lag with shielding improves TIR by $+4.46\%$ but increases CV by $+4.39\%$; a similar pattern appears for RCPO and TRPO-Lag, with modest TIR/Risk Index gains but worse CV. In contrast, some methods benefit substantially: under CRPO, shielding increases TIR by $+11.80\%$ while reducing CV by $-7.28\%$.

## 7. Conclusion and Future Work

This work highlights a critical challenge in safe RL: training-time constraint satisfaction does not consistently translate to reliable performance under distribution shift. By introducing a unified clinical simulator and an OOD safety benchmark, we empirically identify a *safety generalization gap* across eight commonly used safe RL algorithms, where policies verified as safe on training patients frequently violate

| Algorithm | TIR $\uparrow$ (%) | $\Delta$TIR $\uparrow$ | Risk Index $\downarrow$ | $\Delta$Risk $\downarrow$ | CV $\downarrow$ (%) | $\Delta$CV $\downarrow$ |
|---|---|---|---|---|---|---|
| CPO | $85.50 \pm 1.65$ | +4.70 | $3.44 \pm 0.31$ | -1.51 | $25.40 \pm 1.08$ | -2.56 |
| CUP | $73.33 \pm 4.08$ | -0.22 | $7.14 \pm 1.12$ | -0.04 | $35.93 \pm 2.70$ | -0.72 |
| FOCOPS | $82.63 \pm 2.27$ | +3.07 | $4.53 \pm 0.42$ | -0.97 | $29.63 \pm 3.52$ | -4.26 |
| CRPO | $83.76 \pm 2.67$ | +7.83 | $3.92 \pm 0.57$ | -2.16 | $28.77 \pm 1.03$ | -1.16 |
| PCPO | $35.12 \pm 1.25$ | -0.14 | $21.30 \pm 0.75$ | +0.19 | $46.40 \pm 0.86$ | -3.58 |
| PPOLag | $85.59 \pm 0.99$ | +8.05 | $3.96 \pm 0.22$ | -1.79 | $29.18 \pm 0.16$ | -3.21 |
| RCPO | $86.45 \pm 0.58$ | +6.90 | $3.34 \pm 0.04$ | -2.26 | $26.22 \pm 1.24$ | -3.92 |
| TRPOLag | $82.70 \pm 2.16$ | +5.79 | $4.67 \pm 0.67$ | -1.35 | $31.29 \pm 0.74$ | -0.35 |

*Table 3.* Effect of predictive shielding on T1D control. Shielding improves all metrics across the all baselines. The exceptions are CUP and PCPO, where shielding leads to slight reductions in TIR and Risk Index, though it still improves CV.

| Algorithm | TIR $\uparrow$ (%) | $\Delta$TIR $\uparrow$ | Risk Index $\downarrow$ | $\Delta$Risk $\downarrow$ | CV $\downarrow$ (%) | $\Delta$CV $\downarrow$ |
|---|---|---|---|---|---|---|
| CPO | $95.05 \pm 1.41$ | +13.54 | $2.06 \pm 0.34$ | -2.82 | $19.97 \pm 1.97$ | -6.68 |
| CUP | $80.21 \pm 2.08$ | +4.22 | $5.26 \pm 0.54$ | -0.89 | $34.73 \pm 2.74$ | +0.53 |
| FOCOPS | $76.65 \pm 0.76$ | +3.48 | $5.88 \pm 0.37$ | -0.90 | $33.89 \pm 1.27$ | -0.40 |
| CRPO | $86.50 \pm 7.69$ | +7.96 | $4.27 \pm 1.94$ | -1.04 | $28.59 \pm 7.03$ | +0.04 |
| PCPO | $43.63 \pm 1.34$ | +2.00 | $17.84 \pm 0.46$ | -0.73 | $51.09 \pm 0.72$ | -0.09 |
| PPOLag | $92.95 \pm 2.73$ | +14.15 | $2.70 \pm 0.65$ | -2.38 | $25.03 \pm 2.97$ | -5.50 |
| RCPO | $84.28 \pm 10.45$ | +12.73 | $4.55 \pm 2.50$ | -2.41 | $30.12 \pm 9.66$ | -0.75 |
| TRPOLag | $81.37 \pm 5.33$ | +4.13 | $4.56 \pm 1.15$ | -1.34 | $27.43 \pm 1.74$ | -0.75 |

*Table 4.* Effect of predictive shielding on T2D control. Shielding consistently improves TIR and Risk Index across all algorithms. However, CUP and CRPO show worsened CV.

clinical constraints on unseen individuals.

To address this gap, we introduce test-time predictive shielding, an algorithm-agnostic framework that enables zero-shot safety generalization by verifying actions through forward simulation. By using Basis-Adaptive Neural ODEs for precise patient-specific forecasting, this shielding mechanism consistently improves clinical safety across diverse populations where static heuristics fail.

**Limitations and future work.** Our framework focuses on physiological distribution shifts that are partially inferable from recent context. Extreme events such as unannounced meals or acute stress may invalidate the shield's dynamics forecasts. Moreover, our evaluation is studied in a simulated clinical environment; extending the benchmark and shielding to real-world deployment requires validation under real or hybrid clinical data streams and prospective clinical protocols. Future work should incorporate explicit epistemic uncertainty, explore tighter integration between runtime shielding and policy learning, and study sim-to-real transfer under clinician-in-the-loop. By releasing our unified clinical simulator and benchmark, we provide a reusable platform for studying safety under distribution shift.

## Acknowledgments

This work was supported in part by the U.S. National Science Foundation grant CCF-1942836.

| Algorithm | TIR ↑ (%) | ΔTIR ↑ | Risk Index ↓ | ΔRisk ↓ | CV ↓ (%) | ΔCV ↓ |
|---|---|---|---|---|---|---|
| CPO | 80.02 ± 5.53 | +5.17 | 5.20 ± 1.66 | -1.17 | 26.43 ± 5.71 | -2.83 |
| CUP | 84.49 ± 5.82 | +3.66 | 4.46 ± 1.38 | -0.68 | 25.98 ± 3.28 | -3.85 |
| FOCOPS | 79.68 ± 1.43 | +5.44 | 5.10 ± 0.45 | -1.41 | 33.15 ± 1.01 | -1.53 |
| CRPO | 85.55 ± 0.00 | +11.80 | 3.54 ± 0.00 | -2.52 | 20.72 ± 0.00 | -7.28 |
| PCPO | 45.05 ± 0.21 | +2.92 | 17.84 ± 0.20 | -1.29 | 51.22 ± 0.14 | -0.49 |
| PPOLag | 81.75 ± 3.05 | +4.46 | 4.91 ± 0.81 | -1.08 | 33.33 ± 3.23 | +4.39 |
| RCPO | 80.32 ± 3.54 | +0.93 | 5.11 ± 1.07 | -0.07 | 30.48 ± 4.18 | +4.95 |
| TRPOLag | 73.37 ± 0.97 | +0.67 | 6.56 ± 0.18 | -0.77 | 36.39 ± 0.40 | +5.49 |

*Table 5.* Effect of predictive shielding on T2D (without an insulin pump) control. While shielding consistently improves TIR and Risk Index, it degrades CV for some algorithms. This divergence highlights the importance of using multidimensional metrics to detect instability.

## Impact Statement

This work studies whether training-time safety guarantees in safe RL transfer to test-time safety during deployment under distribution shift. We use diabetes management as a safety-critical testbed. We introduce a unified clinical simulator for OOD benchmark and study test-time predictive shielding as an algorithm-agnostic runtime mechanism for filtering unsafe actions using learned dynamics forecasting.

**Applications**. Our benchmark can standardize evaluation of Safe RL under realistic physiological variability, and predictive shielding can serve as a deployment-time safety layer for sequential decision systems without retraining base controllers.

**Implications**. Potential benefits include making safety failures under distribution shift more visible, encouraging more cautious claims about safety, and improving research norms for out-of-distribution evaluation. Risks include (i) over-reliance on runtime shields if dynamics models fail under unmodeled disturbances, (ii) benchmark overfitting that prioritizes simulator metrics over real-world safety. Follow-on work using real patient data may raise privacy and governance concerns.

**Initiatives**. We encourage (i) broader stress testing beyond parametric shifts (noise, rare events), (ii) uncertainty-aware shielding with other safety logic, (iii) privacy- and reproducibility-focused norms for any work involving real patient data.

Overall, this work aims to improve how the community measures and mitigates safety under distribution shift. This is valuable if verified properly, but dangerous if users rely on it too much without through verification.

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

# A. Simulator

## A.1. Physiological ODE Dynamics for T1D and T2D

We implement a differentiable glucose–insulin simulation environment in JAX (Bradbury et al., 2018), grounded in the compartmental ordinary differential equations (ODEs) of the established UVA/Padova simulator lineage (Man et al., 2014; Visentin et al., 2018) and enhanced with control-oriented pharmacodynamics (Hovorka et al., 2004). This unified framework supports both Type 1 Diabetes (T1D) and a hybrid Type 2 Diabetes (T2D) formulation characterized by endogenous insulin secretion and resistance, consistent with the Padova T2D population models (Visentin et al., 2020). To bridge the gap between idealized dynamics and real-world patient variability, we augment the core physiology with three critical extensions: (i) heart-rate-driven physical activity dynamics (Dalla Man et al., 2009), (ii) circadian modulation of metabolic parameters (Visentin et al., 2015), and (iii) structured process and sensor noise (Man et al., 2014). The continuous system dynamics are integrated at one-minute resolution, enabling high-fidelity in silico evaluation of glucose control algorithms. The constitutive subsystems are detailed below.

**Model State Vector and Inputs.** The full ordinary differential equation (ODE) state vector for the hybrid Type 1/ Type 2 diabetes simulator comprises 18 states, integrating subsystems for meal absorption, glucose kinetics, insulin pharmacokinetics/pharmacodynamics, exercise effects, subcutaneous glucose filtering (CGM proxy), and optional endogenous secretion:

$$x = \begin{bmatrix} D_1 & D_2 & D_3 & G_p & G_t & x_1 & x_2 & x_3 \\ I_p & I_l & I_{sc1} & I_{sc2} & G_{sc} & E_1 & T_E & E_2 \end{bmatrix}.$$

The states and their corresponding units are defined as follows:

- **Gut Subsystem**: $D_1, D_2, D_3$ (solid and liquid stomach, and intestine) are in absolute **mg** to ensure mass conservation for meal inputs (Man et al., 2014).

- **Glucose Subsystem**: $G_p$ (plasma), $G_t$ (tissue), and $G_{sc}$ (subcutaneous proxy) are normalized in mg/kg (Man et al., 2014).

- **Insulin PK**: $I_{sc1}, I_{sc2}$ (subcutaneous compartments), $I_p$ (plasma), and $I_l$ (liver/portal) are in pmol/kg (Man et al., 2014)

- **Insulin PD**: $x_1, x_2, x_3$ are parallel remote action states (unitless or scaled sensitivities), primarily active in Type 2 diabetes or hybrid configurations (Dalla Man et al., 2007; Visentin et al., 2020).

- **Physical Activity**: $E_1$ (filtered heart rate excess), $T_E$ (dynamic time constant), and $E_2$ (accumulated intensity) capture exercise dynamics driven by heart rate (Dalla Man et al., 2009).

- **Secretion**: $Y$ (dynamic secretion drive in mU/min) and $G_f$ (filtered glucose in mmol/L) are enabled only for scenarios with residual beta-cell function (e.g., T2D or honeymoon T1D) (Dalla Man et al., 2007).

The system is driven by the control input vector $u(t)$ (applied per simulation minute):

$$u = \begin{bmatrix} \text{CHO}_t & \text{Ins}_t \end{bmatrix}^\top,$$

where $\text{CHO}_t$ is carbohydrate intake (g/min), $\text{Ins}_t$ is exogenous insulin delivery (U/min).

This composite structure allows for seamless switching between strict Type 1 scenarios (zero secretion, direct insulin actions) and Type 2/hybrid scenarios while maintaining dimensional consistency and parameter identifiability.

**Non-negativity guard.** To enforce physiological non-negativity consistent with UVA/Padova, we clamp derivatives that would drive a non-positive state further negative:

$$\text{NN}(x_i, \dot{x}_i) = \begin{cases} 0, & x_i \leq 0 \wedge \dot{x}_i < 0, \\ \dot{x}_i, & \text{otherwise}. \end{cases}$$

This guard is applied to selected states (e.g., $G_p, G_t, I_p, I_l, I_{sc1}, I_{sc2}, G_{sc}$).

A.1.1. T1D DYNAMICS $\dot{x} = f_{\text{T1D}}(x, u)$

**Gut subsystem (meal appearance).** Gut subsystem (meal appearance) Postprandial glucose excursions are driven by delayed and nonlinear gut absorption. We use a three-compartment gut model (two stomach, one intestine) with nonlinear gastric emptying, consistent with the foundational UVA/Padova model (Dalla Man et al., 2007; Man et al., 2014) and its extension to single-day simulations (Visentin et al., 2018). Carbohydrates enter the solid stomach compartment $D_1$ as an impulse-like input (mass conservation), with $\text{CHO}_t$ converted to mg/min:

$$m_t = 1000\,\text{CHO}_t.$$

Let $q_{\text{sto}} = D_1 + D_2$ represent total stomach content. Following the UVA/Padova logic, gastric emptying is governed by a nonlinear rate $k_{gut}(q_{sto}; D_{\text{bar}})$, which depends on the total ingested glucose $D_{\text{bar}}$ to account for stacking effects from prior meals (Visentin et al., 2018). Here, $D_{\text{bar}} = Q_{\text{sto}}^{\text{last}} + 1000\,\text{food}^{\text{last}}$. The system dynamics are:

$$\dot{D}_1 = -k_{gri}D_1 + m_t,$$
$$\dot{D}_2 = k_{gri}D_1 - k_{gut}(q_{sto}; D_{\text{bar}})D_2,$$
$$\dot{D}_3 = k_{gut}(q_{sto}; D_{\text{bar}})D_2 - k_{abs}D_3,$$

where $k_{gri}$ is the constant grinding rate, $k_{gut}$ varies between $k_{\min}$ and $k_{\max}$ based on stomach fullness, and $k_{abs}$ is the constant rate of intestinal absorption. The rate of glucose appearance in plasma is:

$$R_{-}a = \frac{f \cdot k_{abs} \cdot D_{-}3}{BW} (\text{mg/kg/min}).$$

**Exercise states and glucose shunts.** The effects of physical activity on glucose dynamics are modeled building upon the foundational framework of Breton (2008) and Dalla Man et al. (2009), which established heart rate as a reliable, non-invasive proxy for exercise intensity and introduced dual-timescale dynamics for immediate and prolonged effects on insulin-independent glucose disposal. With resting heart rate $HR_0$ and maximum heart rate $HR_{\max} = 220 - \text{age}$ (using the standard Fox formula (Fox & Naughton, 1972)), the instantaneous heart rate is normalized as

$$HR = \text{HRr}_t (HR_{\max} - HR_0) + HR_0,$$

where $\text{HRr}_t \in [0, 1]$ represents relative intensity. This normalization facilitates parameter identifiability across individuals.

We extend this framework with enhanced multi-timescale dynamics to capture variable decay rates, including a dynamic time constant for more realistic post-exercise tails during intermittent or repeated bouts:

$$\dot{E}_1 = \frac{HR - HR_0 - E_1}{\tau_{HR}},$$

$$f_{E1} = \frac{z}{1+z}, \quad z = \left(\frac{E_1}{\alpha_{HR} HR_0}\right)^n,$$

$$\dot{T}_E = \frac{c_1 f_{E1} + c_2 - T_E}{\tau_{ex}},$$

$$\dot{E}_2 = -\left(\frac{f_{E1}}{\tau_{in}} + \frac{1}{T_E}\right) E_2 + \frac{f_{E1} T_E}{c_1 + c_2}.$$

Here, $E_1$ represents low-pass filtered excess heart rate (time constant $\tau_{HR}$), smoothing rapid changes and modeling delayed onset of metabolic effects (Dalla Man et al., 2009). The Hill-type sigmoidal function $f_{E1}$ (with threshold $\alpha_{HR}$ and steepness $n$) introduces a physiological activation threshold, beyond which effects amplify nonlinearly, consistent with prior models (Dalla Man et al., 2009). Unlike the fixed time constants in the original framework (Dalla Man et al., 2009), the auxiliary state $T_E$ and $E_2$ dynamics enable rapid accumulation during intense activity (fast charging via $f_{E1}$) and variable decay accelerated offset during exercise but prolonged post-exercise tail (slow decay governed by $T_E$).

Exercise induces additional insulin-independent glucose disposal via (i) a mild, rapid component proportional to intensity and (ii) a stronger, accumulated saturating component

with safeguards:

$$Q_{E1} = \beta_{ex} \frac{E_1}{HR_0},$$

$$V_{\max}^{ex} = \alpha_{QE} E_2^2, \quad s_p = \frac{V_{\max}^{ex} G_p}{K_m^{ex} + G_p}, \quad s_t = \frac{V_{\max}^{ex} G_t}{K_m^{ex} + G_t},$$

$$Q_{E21} = \min(s_p, c_{cap} G_p), \quad Q_{E22} = \min(s_t, c_{cap} G_t)$$

The immediate term $Q_{E1}$ captures quick muscle contraction-driven uptake (Dalla Man et al., 2009).

Furthermore, unlike the linear relationships in \ (Dalla Man et al., 2009), the prolonged term uses quadratic scaling in $E_2$ to better capture super-linear responses to accumulated exercise dose, while MichaelisMenten kinetics reflect transporter-limited (GLUT4-mediated) saturation (Bizzotto et al., 2016). Explicit per-minute caps ($c_{\text{cap}}$) bound fractional disposal rates, preventing unrealistically aggressive hypoglycemia in simulations-a practical enhancement for numerical stability and alignment with perfusion/delivery constraints observed physiologically (Frank et al., 2021).

These extensions improve fidelity for scenarios involving variable-intensity or prolonged activity, while preserving the core physiological insights from the foundational works.

**Glucose transport, utilization, and production.** We use a two-compartment glucose kinetics model with endogenous glucose production (EGP), insulin-dependent utilization, and renal excretion, consistent with the foundational equations of the UVA/Padova Type 1 Diabetes Simulator framework (Dalla Man et al., 2007; Man et al., 2014). The model tracks glucose mass in the plasma ($G_p$) and tissue ($G_t$) compartments (mg/kg). Let $I$ denote the delayed insulin action (pmol/L). Endogenous glucose production is modeled as a linearly suppressed function of glucose and insulin, constrained to be non-negative:

$$EGP = \max(k_{p1} - k_{p2} G_p - k_{p3} I, 0).$$

Renal excretion ($E$) occurs via a threshold-linear relationship when plasma glucose exceeds $k_{e2}$:

$$E = \begin{cases} k_{e1}(G_p - k_{e2}) & \text{if } G_p > k_{e2}, \\ 0 & \text{otherwise}. \end{cases}$$

Insulin-dependent utilization ($U_{id}$) in the tissue compartment follows Michaelis-Menten kinetics, where the maximum rate $V_m$ is linearly modulated by insulin:

$$V_m = V_{m0} + V_{mx} I, \quad U_{id} = \frac{V_m G_t}{K_{m0} + G_t}.$$

The system dynamics are governed by the following coupled differential equations:

$$\dot{G}_p = EGP + R_a - F_{snc} - E - k_1 G_p + k_2 G_t - Q_{E21},$$

$$\dot{G}_t = k_1 G_p - k_2 G_t - U_{id} - Q_{E1} - Q_{E22}.$$

Here, $F_{\text{snc}}$ represents constant insulin-independent glucose uptake by the brain and central nervous system. To ensure mass conservation while modeling metabolic consumption, the exercise-induced fluxes ($Q_{E*}$) are treated as net disposal terms. $Q_{E21}$ represents rapid, intensity-driven uptake directly from the plasma compartment, while $Q_{E1}$ and $Q_{E22}$ represent sustained uptake from the tissue compartment.

**Insulin Absorption, Kinetics, and Action Delays.** We model insulin pharmacokinetics using the subsystem of the UVA/Padova Type 1 Diabetes Simulator (Man et al., 2014; Kovatchev et al., 2009), which captures the specific dissociation kinetics of rapid-acting insulin analogs. External insulin infusion $\text{Ins}_t(\text{U/min})$ is converted to a mass flux IIR (pmol/kg/min) entering the subcutaneous space. The system tracks two subcutaneous compartments ($I_{sc1}, I_{sc2}$) and the physiological recycling between plasma ($I_p$) and liver ($I_l$):

$$
\begin{aligned}
\dot{I}_{sc1} &= -(k_{a1} + k_d)\, I_{sc1} + IIR \\
\dot{I}_{sc2} &= k_d I_{sc1} - k_{a2} I_{sc2} \\
\dot{I}_p &= -(m_2 + m_4)\, I_p + m_1 I_l + k_{a1} I_{sc1} + k_{a2} I_{sc2} \\
\dot{I}_l &= -(m_1 + m_{30})\, I_l + m_2 I_p
\end{aligned}
$$

All state variables are constrained to be non-negative. Plasma insulin concentration is given by $I = I_p/V_I$ (pmol/L). To capture the differential time-lags of insulin action on peripheral utilization and hepatic production, we extend the model with remote action states, a structure inspired by Hovorka et al. (2004):

$$
\dot{x}_1 = -p_{2u} x_1 + p_{2u}\,(I - I_b), \quad \dot{x}_2 = -k_i\,(x_2 - I),
$$
$$
\dot{x}_3 = -k_i\,(x_3 - x_2).
$$

Here, $x_1$ represents the delayed insulin action on glucose utilization (consistent with the standard minimal model), while the chained states $x_2$ and $x_3$ provide additional delays for hepatic suppression or other slow-acting dynamics.

### A.1.2. T2D HYBRID DYNAMICS $\dot{x} = f_{\text{T2D}}(x, u)$

T2D model retains the same gut, exercise, glucose transport, and exogenous insulin kinetics, but adds endogenous insulin secretion and uses a T2D-style EGP suppression driven by an insulin-effect state $x_3$.

**Endogenous secretion block.** To model residual beta-cell function for type 2 diabetes scenarios, we extend the simulator with the endogenous secretion subsystem established in the foundational normal physiology model (Dalla Man et al., 2007) and adapted for impaired secretion in the Padova Type 2 Diabetes Simulator (Visentin et al., 2020).

Plasma glucose concentration $G_p(\text{mg/dL})$ is converted to molar concentration $G(\text{mmol/L})$ using the standard factor of 18. The secretion dynamics are driven by a filtered glucose signal $G_f$ and a dynamic control variable $Y$:

$$
\dot{G}_f = \frac{G - G_f}{\tau_{dG}},
$$
$$
\dot{Y} = -\alpha_s Y + \beta_s \max(G - h, 0) + K_{\text{deriv}} \max\left(\dot{G}_f, 0\right).
$$

Here, $Y$ captures the dynamic beta-cell response, including static responsivity to glucose above a threshold $h$ and rate-sensitive potentiation via the derivative term $\dot{G}_f$. This structure allows for modeling impaired function for T2D by adjusting sensitivity parameters ($\beta_s, K_{\text{deriv}}$) and thresholds ($h$) (Visentin et al., 2020).

Total secretion combines a weight-scaled basal rate with this dynamic drive:

$$
S_{\text{basal}} = S_b^{(kg)} BW, \quad S_{\text{total}} = S_{\text{basal}} + \beta_{\text{cell}} Y.
$$

This total secretion is converted to a portal insulin mass flux $S_{\text{endog}}$ (pmol/kg/min) using the unit conversion factor $f_{\text{conv}} \approx 6$ (mU to pmol):

$$
S_{\text{endog}} = \frac{f_{\text{conv}}\, S_{\text{total}}}{BW}.
$$

This endogenous flux enters the liver insulin compartment, modifying the standard insulin kinetics to account for portal delivery and hepatic first-pass extraction:

$$
\dot{I}_l = -(m_1 + m_{30})\, I_l + m_2 I_p + S_{\text{endog}}
$$

**Insulin Action and Resistance (T2D Effects).** To model the insulin resistance and differential pharmacodynamics characteristic of Type 2 Diabetes, we employ the parallel remote action subsystem from the Padova T2D Simulator framework (Visentin et al., 2020; Dalla Man et al., 2007). This approach captures the distinct timescales of insulin action on different tissues (e.g., rapid peripheral uptake vs. slower hepatic suppression) using three parallel first-order effect compartments.

First, plasma insulin is converted to standard clinical units (mU/L) to align with literature-derived sensitivity parameters:

$$
I_{\text{mU/L}} = \frac{I_p/V_i}{6}.
$$

The remote insulin action states ($x_1, x_2, x_3$) evolve according to:

$$
\begin{aligned}
\dot{x}_1 &= -k_{a1} x_1 + S_{I1} I_{\text{mU/L}}, \\
\dot{x}_2 &= -k_{a2} x_2 + S_{I2} I_{\text{mU/L}}, \\
\dot{x}_3 &= -k_{a3} x_3 + S_{I3} I_{\text{mU/L}}.
\end{aligned}
$$

Here, $k_{ai}$ represents the rate of onset/offset for each action, and $S_{Ii}$ represents insulin sensitivity. In T2D simulations,

resistance is modeled by reducing the sensitivity parameters ($S_{Ii}$) compared to healthy controls (Visentin et al., 2020). Unlike models that explicitly subtract basal insulin, this formulation embeds basal action within the parameters.

These states modulate the glucose subsystem defined previously: $x_1$ and $x_2$ typically drive peripheral utilization ($V_m$), while the slower state $x_3$ drives the suppression of endogenous glucose production (EGP).

**T2D Endogenous Glucose Production and System Dynamics.** In Type 2 diabetes modeling, we scale basal endogenous glucose production $EGP_0$ (mmol/min whole-body) and central nervous system utilization $F_{cns0}$(mmol/min) to mass-specific units using the molecular weight of glucose (180 g/mol):

$$EGP_0^{\mathrm{mg/kg/min}} = \frac{180 \cdot EGP_0}{BW}, \quad F_{cns} = \frac{180 \cdot F_{cns0}}{BW}.$$

To reflect hepatic insulin resistance, suppression is mediated by the remote action state $x_3$, clipped to a maximum effect of 95% to model residual production (Visentin et al., 2020):

$$
\begin{aligned}
x_3^{eff} &= \min\left(\max\left(x_3, 0\right), 0.95\right), \\
EGP &= \max\left(EGP_0^{\mathrm{mg/kg/min}}\left(1 - x_3^{eff}\right), 0\right).
\end{aligned}
$$

Renal excretion $E$ and insulin-dependent utilization $U_{id}$ (modulated via remote actions $x_1/x_2$) follow established formulations. Exercise-induced disposal shunts ($Q_{E_*}$) are treated as net removal terms to preserve mass balance. The two-compartment glucose dynamics (plasma $G_p$ and tissue $G_t$, mg/kg) are:

$$
\begin{aligned}
\dot{G}_p &= EGP + R_a - F_{cns} - E - k_1 G_p + k_2 G_t - Q_{E21}, \\
\dot{G}_t &= k_1 G_p - k_2 G_t - U_{id} - Q_{E22} - Q_{E1}.
\end{aligned}
$$

(Note: All rates are clipped to maintain non-negative states). Exogenous insulin absorption and plasma kinetics ($I_{sc1}, I_{sc2}, I_p$) are identical to the T1D model (Man et al., 2014), while the liver compartment is augmented by the portal endogenous secretion $S_{\mathrm{endog}}$ defined previously. A subcutaneous glucose filter for the CGM proxy $\dot{G}_{sc}$ matches standard implementations.

### A.1.3. NUMERICAL INTEGRATION AND REALISM WRAPPERS

**RK4 integration with one-minute mini-steps.** For both T1D and T2D, we integrate the ODE with RK4:

$$x_{t+\Delta t} = x_t + \frac{\Delta t}{6}\left(k_1 + 2k_2 + 2k_3 + k_4\right),$$

where $k_1 = f\left(x_t, u_t\right), k_2 = f\left(x_t + \frac{\Delta t}{2} k_1, u_t\right), k_3 = f\left(x_t + \frac{\Delta t}{2} k_2, u_t\right), k_4 = f\left(x_t + \Delta t k_3, u_t\right)$. After each step, we project selected states to enforce non-negativity and clip $T_E$ to a valid range.

**Uncertainty, Circadian Modulation, and Sensor Noise.** To enhance realism in multi-day simulations, the model incorporates physiological and practical variabilities extended from the core equations. Before numerical integration, user-specified actions are perturbed to mimic real-world uncertainties: insulin delivery is subject to random mechanical errors (e.g., pump inaccuracies), and announced carbohydrate intakes are perturbed by misestimation factors (e.g., $\pm 20\%$), consistent with standard challenging scenarios in FDA-accepted in silico trials (Man et al., 2014; Visentin et al., 2018).

Circadian rhythmicity is modeled via a time-of-day-dependent factor $c(t)$ (e.g., a periodic function peaking in the early morning to capture the dawn phenomenon). This modulation is applied additively to the glucose dynamics to represent diurnal variations in hepatic glucose output and insulin sensitivity (Visentin et al., 2015). For Type 1 Diabetes, the constant EGP parameter $k_{p1}$ is modulated:

$$R_{\mathrm{circ}}(t) = (c(t) - 1)k_{p1}.$$

For Type 2 Diabetes, the modulation scales with the current suppressed production rate:

$$R_{\mathrm{circ}}(t) = (c(t) - 1)EGP_0^{\mathrm{mg}}\left(1 - x_3^{\mathrm{eff}}\right).$$

This term is added to the plasma glucose derivative: $\dot{G}_p = \cdots + R_{\mathrm{circ}}(t)$.

Finally, structured temporally correlated process noise-modeled via an Ornstein-Uhlenbeck (OU) process for mean-reverting fluctuations-is injected into glucose-related states to capture unmodeled physiological dynamics (e.g., stress or activity residuals). CGM readings are derived from the subcutaneous glucose state with added colored measurement noise and transduction lag, aligning with validated sensor error models (Man et al., 2014).

### A.2. Virtual Patient Simulation

#### A.2.1. VIRTUAL PATIENT POPULATION

We employ a cohort of 30 in silico subjects adapted from the Simglucose library (Xie, 2018), which implements the FDA-accepted UVA/Padova Type 1 Diabetes Simulator lineage (Kovatchev et al., 2009; Man et al., 2014). To capture realistic physiological variability, the cohort is stratified evenly across three age groups—Children, Adolescents, and Adults (10 subjects each). Figure 3 summarizes the cohort-level distributions of body weight and basal physiological parameters for the virtual patients.

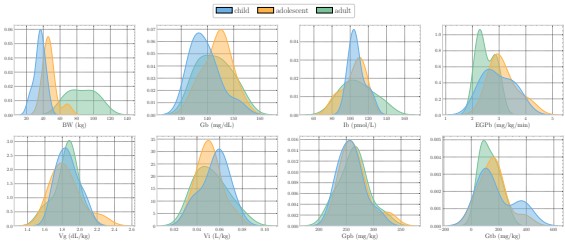

*Figure 3.* **Patient Parameters**: Kernel density estimates of key virtual-patient parameters across child, adolescent, and adult cohorts. The distributions reflect cohort-level morphology and basal physiology, including body weight, basal glucose and insulin concentrations, endogenous glucose production, and glucose/insulin distribution volumes, highlighting age-dependent shifts in metabolic state.

**Cohort Demographics and Basal Physiology.** Each virtual patient is parameterized using morphological and metabolic profiles drawn from validated clinical cohorts, spanning pediatric to adult populations.

- **Body Weight:** The cohort covers a broad range of body masses: $23.7$–$45.5\,\mathrm{kg}$ (mean $35.9\,\mathrm{kg}$) for children, $37.9$–$68.7\,\mathrm{kg}$ (mean $47.7\,\mathrm{kg}$) for adolescents, and $63.0$–$111.1\,\mathrm{kg}$ (mean $86.1\,\mathrm{kg}$) for adults.

- **Basal States:** Age-dependent steady-state regulation is reflected in basal plasma glucose and insulin levels. Mean basal glucose ($G_b$) is approximately 139, 144, and $143\,\mathrm{mg/dL}$ for children, adolescents, and adults, respectively, while corresponding basal insulin concentrations ($I_b$) are approximately 107, 104, and $108\,\mathrm{pmol/L}$.

- **Kinetics:** Basal endogenous glucose production ($\mathrm{EGP}_b$) is highest in children ($\sim 2.96\,\mathrm{mg/kg/min}$) and decreases with age, reaching approximately $2.51\,\mathrm{mg/kg/min}$ in adults. Distribution volumes are tightly clustered across cohorts, with glucose volume $V_g \approx 1.85$–$1.86\,\mathrm{dL/kg}$ and insulin volume $V_i \approx 0.051$–$0.056\,\mathrm{L/kg}$.

Core physiological parameters include gut absorption ($k_{\mathrm{abs}}$), subcutaneous insulin kinetics ($k_{sc}, k_d$), renal thresholds ($k_{e1,2}$), and hepatic extraction ($HE_b$). These are loaded from standardized configuration files to ensure unit consistency (masses in mg/kg, insulin in pmol/kg).

**Parameter Adjustments for Diabetes Types.** Virtual patient parameters are initialized from the baseline CSV physiology, subject to global pharmacodynamic tuning (scaling $V_g, G_{pb}$, and $G_{tb}$ by 0.65 for sharper excursions). Disease specific adaptations are then applied independently, ensuring physiological consistency without cross-contamination between Type 1 (T1D) and Type 2 (T2D) pipelines.

**Type 1 Diabetes (T1D)**: Aligning with the UVA/Padova simulator's zero-secretion assumptions (Kovatchev et al., 2009; Man et al., 2014), this adaptation enforces:

1. $\beta$-cell Failure: Residual function is set to zero ($S_b = 0$) with no insulin resistance adjustment (factor 1.0).

2. Therapy: Continuous subcutaneous insulin infusion (pump) is mandated with a basal rate of $\sim 0.011 \times \mathrm{BW}$ U/h.

3. Pharmacodynamics: To ensure challenging control scenarios, carbohydrate absorption rates ($k_{\max}, k_{abs}$) are doubled ($2.0\times$), and insulin sensitivity ($V_{mx}$) is scaled by 0.8.

4. Stability: An auto-balancing routine recomputes the insulin action slope ($V_{mx}$) and EGP offset ($k_{p1}$) to enforce steady-state equilibrium at the reported basal masses.

**Type 2 Diabetes (T2D)**: T2D adaptations bypass T1D steps, deriving remote-action and secretion parameters directly from the baseline to capture impaired sensitivity and compensatory hyperinsulinemia, inspired by established meal models (Dalla Man et al., 2007; Visentin et al., 2020). The physiology is upscaled (BW $\times 1.15$, insulin pools $\times 1.25$, hepatic extraction $\times 0.85$) and split into two modalities:

1. T2D with Pump: Retains residual $\beta$-cell function ($\sim 25\%$) and applies an insulin resistance factor of 2.5. The system computes a basal infusion rate that balances total clearance against endogenous secretion.

2. T2D without Pump: Disables exogenous basal delivery ($0\,\mathrm{U/h}$). To compensate, residual $\beta$-cell function is increased to $\sim 30\%$, insulin resistance to 2.8, and rate-sensitive secretion gain ($K_{\mathrm{deriv}}$) is set to 30.0 .

### A.2.2. BEHAVIORAL ADHERENCE AND ACTION ACCEPTANCE LOGIC

To bridge the gap between algorithmic control and realistic patient behavior, agent recommendations are not executed directly on the physiological model. Instead, they pass through a hierarchical acceptance logic layer that simulates non-compliance, physiological safety constraints, and mutual exclusion between conflicting activities.

**Meal Acceptance and Rescue Override.** Carbohydrate ingestion is modeled as a conditional process. A recommended meal is executed only if the following constraints are met:

1. **Adherence & Timing:** To isolate control policy performance, we assume perfect patient compliance (probability $p = 1.0$). However, a temporal refractory period

(default: 60 min) must have elapsed since the previous meal to prevent non-physiological eating frequencies.

2. **Hyperglycemic Safety:** Meal requests are blocked if the current Continuous Glucose Monitor (CGM) reading exceeds $200\,\mathrm{mg/dL}$, preventing exacerbation of existing hyperglycemia.

3. **Mutual Exclusion:** Meal ingestion is strictly mutually exclusive with exercise; a patient cannot eat while the exercise state is active.

4. **Hypoglycemia Override:** To model critical self-preservation behavior consistent with clinical guidelines (Bajaj et al., 2025), the acceptance logic forces the execution of a meal request, bypassing all safety blocks (including time windows), if the patient is hypoglycemic (CGM $< 70\,\mathrm{mg/dL}$).

**Bolus Safety Gates.** Insulin bolus recommendations are subject to similar safety gating to prevent iatrogenic events:

1. **Hypoglycemic Safety:** Boluses are strictly rejected if the CGM reading is below the clinical hypoglycemia threshold of $70\,\mathrm{mg/dL}$ (Bajaj et al., 2025).

2. **Frequency Constraints:** To avoid "insulin stacking," boluses are blocked if the time elapsed since the last injection is within the defined safe window.

3. **Daily Limits:** Both meals and boluses are subject to a maximum daily count to prevent non-physiological frequency of interventions.

**Actuation and Uncertainty.** Once an action is accepted, it is converted from a normalized control signal into a physiological quantity scaled by patient-specific maxima. To simulate estimation error (e.g., carb counting inaccuracy), we inject Gaussian noise into the final quantity:

$$A_{\text{executed}} = \max\left(0, A_{\text{target}} \cdot \left(1 + \mathcal{N}\left(0, \sigma^2\right)\right)\right)$$

We apply $\sigma = 0.10$ ( $10\%$ error) for carbohydrate estimation and a tighter $\sigma = 0.01$ ( $1\%$ error) for insulin dosing. The final carbohydrate mass is not absorbed instantly but is fed into the gut compartment linearly over time via a rate-limited process (default $10\,\mathrm{g/min}$ ), while insulin is injected immediately into the subcutaneous depot.

### A.3. Observation Interface

At each control step the simulator constructs a deterministic 14-dimensional observation vector

$$o_t \in \mathbb{R}^{14}.$$

Each component is derived from the internal simulator state after ODE integration of the previous 5-minute interval. The components, in fixed order, are:

1. **CGM glucose (mg/dL):** The agent observes CGM, a noisy measurement derived from the simulator's latent CGM channel $G_{sc}$, which is a first-order filtered version of plasma glucose $G_p$:

$$\dot{G}_{sc} = \mathrm{NN}(G_{sc}, -k_{sc}G_{sc} + k_{sc}G_p).$$

The observed CGM value is obtained by applying realistic sensor effects to $G_{sc}$, including slow multiplicative bias and scale drift (0.9–1.1), additive Gaussian noise, occasional dropout (forward-filled), and hard clipping to $[40, 400]$ mg/dL. The true plasma glucose $G_p$ is not directly observable by the agent.

2. **Bolus insulin-on-board (IOB)(U):** The remaining active bolus insulin is computed as

$$\mathrm{IOB}_{\text{bolus}}(t) = \sum_{k \in \mathcal{B}_t} b_k\, K_{\text{iob}}(t - k),$$

where $b_k$ (U) is the accepted bolus delivered at time step $k$, $\mathcal{B}_t$ is the set of prior bolus times, and $K_{\text{iob}}$ is a fixed, dimensionless decay kernel modeling insulin activity over time. The resulting $\mathrm{IOB}_{\text{bolus}}(t)$ has units of insulin Units (U). Basal insulin-on-board is intentionally excluded.

3. **Carbohydrates-on-board (COB) (g)**:

$$\mathrm{COB}(t) = \frac{D_1(t) + D_2(t) + S_1(t)}{1000},$$

derived from gut compartments in the digestion model.

4. **CGM trend (mg/dL/min)**: slope between the two most recent CGM readings:

$$\mathrm{trend}(t) = \frac{\mathrm{CGM}(t) - \mathrm{CGM}(t - \Delta t)}{\Delta t}.$$

5-6. **Circadian encodings**: To capture the cyclical nature of daily metabolic rhythms, we project the time of day $t$ (in minutes) onto the unit circle. This results in a two dimensional continuous embedding:

$$\sin\left(\frac{2\pi t}{1440}\right), \quad \cos\left(\frac{2\pi t}{1440}\right)$$

ensuring that the transition from the end of one day to the start of the next is smooth for the policy network.

7-8. **Normalized time since last meal/bolus**: To mitigate the partial observability of physiological states, we explicitly track the temporal proximity of the most

recent intervention events. We define two normalized counters:

$$\text{TSM}(t) = \min\left(\frac{t - t_{\text{meal}}}{180}, 1\right),$$

$$\text{TSB}(t) = \min\left(\frac{t - t_{\text{bolus}}}{180}, 1\right),$$

where $t_{\text{meal}}$ and $t_{\text{bolus}}$ represent the timestamps of the last meal intake and insulin bolus, respectively.

9. **Pending meal size (normalized)**: The quantity of carbohydrates from an ongoing meal that is currently buffered and waiting to be consumed, divided by the maximum meal size and clipped to $[0, 1]$. Unlike instantaneous intake, meals are consumed linearly over time (e.g., 5 g/min); this observation informs the agent of committed but not-yet-absorbed glucose.

10-11. **Daily meal/bolus counts (normalized)**:

$$\text{meal\_count\_norm}\,(t) = \frac{\#\text{ meals today}}{7},$$

$$\text{bolus\_count\_norm}\,(t) = \frac{\#\text{ boluses today}}{8}.$$

These counters track the number of successfully executed interventions within the current 24-hour cycle and reset to zero at midnight. The normalization constants (7 for meals, 8 for boluses) represent hard daily limits; if a counter reaches saturation (1.0), the simulator strictly rejects subsequent requests of that type until the next day.

12-13. **Scheduled-meal look-ahead**: To capture the non-stationary nature of daily life, each patient instance is initialized with a unique, procedurally generated meal schedule at the start of every episode. We provide two look-ahead features to allow the agent to anticipate these varying disturbances:

(a) Normalized time until next scheduled meal: calculated as $\min(\frac{\Delta t_{\text{next}}}{180}, 1)$, where $\Delta t_{\text{next}}$ is the minutes remaining. A value of 0 indicates an imminent meal, while 1 indicates the next meal is $\geq 3$ hours away.

(b) Normalized size of that scheduled meal: $\frac{\text{meal\_size}}{\text{max\_meal}}$, clipped to $[0, 1]$.

14. **Pre-bolus window flag**: binary indicator equal to 1 when the next scheduled meal occurs in 15-30 minutes.

## A.4. Action Interface

At each control step the agent outputs a pair of normalized actions

$$\left(\hat{b}_t, \hat{m}_t\right) \in [0, 1]^2,$$

representing recommended bolus insulin and carb size, respectively. Their physical units are obtained by linear scaling:

$$b_t = \hat{b}_t B_{\text{max}}, \quad m_t = \hat{m}_t M_{\text{max}},$$

where $B_{\text{max}}$ and $M_{\text{max}}$ are patient-specific maximum bolus (units) and maximum meal size (grams). Basal insulin delivery remains exogenous and is applied automatically each minute.

Actions are not guaranteed to execute. The simulator implements clinically motivated acceptance rules, defined in step.py and summarized below.

**Adherence and Constraints.** To focus on the safety and efficacy of the control policy, we assume perfect patient adherence to the recommended action. However, actions are still subject to strict physiological safety constraints:

1. Safety Refractory Period: A 60-minute cooldown is enforced between consecutive meals or boluses.

2. Physiological Guards: Recommendations are automatically rejected if they would exacerbate dangerous states (e.g., meals are blocked if BG > 200 mg/dL; boluses are blocked if BG < 70 mg/dL).

3. Execution Noise: Accepted actions are subject to multiplicative execution noise to simulate metabolic variability: $a_{\text{final}} \sim a_{\text{target}} \cdot \mathcal{N}(1, \sigma^2)$, with $\sigma = 0.1$ for meals and $\sigma = 0.01$ for insulin.

**Spacing Windows.** Meals and boluses must satisfy minimum separation times:

$$t - t_{\text{last meal time}} \geq W_{\text{meal}}, \quad t - t_{\text{last bolus time}} \geq W_{\text{bolus}},$$

where $W_{\text{meal}}$ and $W_{\text{bolus}}$ are patient-specific "safe windows." Actions violating spacing constraints are blocked and reported through the info dictionary.

**CGM-Based Safety Gates.** To prevent clinically unsafe interventions: - Boluses are blocked when CGM < 70mg/dL. - Meals may be blocked when CGM is severely elevated ($\geq 200$mg/dL), reflecting common patient behavior and clinician recommendations.

**Daily Hard Caps.** Each simulated day imposes upper bounds:

$$\#\text{meals/day} \leq 7, \quad \#\text{boluses/day} \leq 8.$$

Attempts beyond the cap are ignored. Counts reset at midnight.

**Interaction with Scheduled Meals.** Scenario meals are exogenous and occur at fixed times unless postponed by a controller-initiated meal within a small safety buffer. Stacking of multiple large meals within short intervals is prevented by temporarily delaying smaller scheduled meals.

**Integration with the Patient ODE.** Accepted boluses and meals are passed directly into the physiological model:

- Bolus insulin is added as an instantaneous dose to the subcutaneous compartment.
- Meal carbohydrates enter the stomach compartment and propagate through the digestion model.
- All effects are integrated at 1 -minute resolution inside the 5 minute control step.

This yields deterministic next-state dynamics given acceptance outcomes and random noise.

### A.5. Reward and Cost Functions

This section provides the complete mathematical definition of the reward and cost functions used in the simulator, including the clinical risk cost, the counterfactual forecasting reward, regularization terms, safety gating, and termination penalties.

**Clinical Risk Cost.** The clinical risk cost models the instantaneous danger of the patient's physiological state based on both blood glucose level $BG$ and the glucose rate of change $\Delta BG$. Thresholds are selected according to standard clinical classifications (Bajaj et al., 2025): mild hypoglycemia below 70 mg/dL, severe below 54 mg/dL, mild hyperglycemia above 180 mg/dL, and severe above 250 mg/dL. The cost is a piecewise combination of mild and severe hypo- and hyperglycemia penalties, together with momentum and low-velocity terms:

$$
\begin{aligned}
C_{\text{risk}}\left(BG, \dot{BG}\right) = {} & W_{\text{hypo,mild}}\left(\frac{\max(0, 70 - BG)}{20}\right) \\
& + W_{\text{hypo,severe}}\left(\frac{\max(0, 54 - BG)}{20}\right)^2 \\
& + W_{\text{hyper,mild}}\left(\frac{\max(0, BG - 180)}{50}\right) \\
& + W_{\text{hyper,severe}}\left(\frac{\max(0, BG - 250)}{50}\right)^2 \\
& + W_{\text{mom}} \cdot \max(0, BG - 160) \cdot \max(0, \Delta BG) \\
& + W_{\text{low-velocity}} \cdot \max(0, -(\Delta BG + 2))
\end{aligned}
$$

Parameter values

$$
\begin{aligned}
W_{\text{hypo, mild}} &= 3.0, & W_{\text{hypo, severe}} &= 10.0 \\
W_{\text{hyper, mild}} &= 1.0, & W_{\text{hyper, severe}} &= 4.0 \\
W_{\text{mom}} &= 0.015, & W_{\text{low-velocity}} &= 0.1
\end{aligned}
$$

The final cost is defined by:

$$
C_{\text{final}} = 0.35 \cdot C_{\text{risk}}.
$$

To avoid penalizing clinically appropriate meals, when a meal is recommended while glucose is within the safe range

$(70 \leq BG \leq 180)$, the risk cost is computed using a baseline continuation (no action) rather than the immediate response.

**Auxiliary Risk Forecast Model.** To compute short-horizon risk estimates for reward shaping, we employ a lightweight differentiable auxiliary risk forecast model. This model is intentionally simpler than the physiologic ODE simulator used for environment transitions and serves only to estimate near-term trends for reward computation.

At each 5-minute step, predicted glucose evolves according to:

$$
BG_{t+1} = \text{clip}\left(BG_t + \Delta_{\text{carb},t} - \Delta_{\text{ins},t} - \Delta_{\text{endo},t}, 40, 600\right).
$$

Each term in the auxiliary forecast model represents the approximate contribution to glucose evolution over a single control interval of $\Delta t = 5$ minutes. These components are designed to capture short-horizon trends relevant for risk estimation, rather than to reproduce the full nonlinear patient dynamics.

- **Carbohydrate Absorption Term** $\triangle_{\text{carb}}$: Carbohydrate effect is computed by convolving the carbohydrate-intake history $H_{\text{carb}}$ with a gammashaped absorption kernel: $\Delta_{\text{carb},t} = (H_{\text{carb}} * K_{\text{carb}})_t \cdot \text{CSF}$. The kernel has the normalized form: $K_{\text{carb}}(\tau) \propto \tau^{\alpha-1} e^{-\tau/\beta}, \tau \geq 0$, and the carbohydrate sensitivity factor is: $\text{CSF} = \phi \cdot \frac{1000}{BW \cdot V_g}$, where $\phi = 0.35$ (absorption efficiency constant), $BW$ is body weight (kg), and $V_g$ is the glucose distribution volume (dL/kg).

- **Insulin Activity Term** $\Delta_{\text{ins}}$: Insulin action is modeled analogously via a convolution of recent insulin doses: $\Delta_{\text{ins},t} = (H_{\text{ins}} * K_{\text{ins}})_t \cdot \text{ISF}$, where ISF is the insulin sensitivity factor and $H_{\text{ins}}$ includes both basal and bolus insulin delivered at each minute. The kernel $K_{\text{ins}}$ is precomputed and represents a standard insulin-activity curve (Walsh/Hovorka-like or gamma-shaped).

- **Endogenous Regulation Term** $\triangle_{\text{endo}}$: Endogenous glucose regulation is modeled as proportional feedback toward a basal glucose setpoint $G_b$ $\Delta_{\text{endo},t} = \beta_{\text{func}} \cdot k_{\text{endo}} \cdot \max(0, BG_t - G_b) \cdot \Delta t$. Here $\beta_{\text{func}} \in [0, 1]$ indicates residual beta-cell function, and $k_{\text{endo}} = 0.02$ is a constant endogenous-regulation rate.

We emphasize that this auxiliary model is used exclusively for short-horizon risk estimation in reward computation.

**Forecasting Procedure for Delta-Risk.** To compute the delta-risk reward, the proxy model is rolled out for a horizon of $H = 24$ steps (2 hours). Two short-horizon trajectories are simulated:

1. **Baseline trajectory** $T_{\text{baseline}}$: No meal or bolus is applied at the current step. Only basal insulin and ongoing absorption terms are propagated.

2. **Action trajectory** $\tau_{\text{action}}$: Uses the agent's proposed meal or bolus at the current step and updates the corresponding carbohydrate and insulin histories.

Each trajectory produces predicted glucose values $\tau^{(k)}$ for $k = 1, \ldots, H$. The delta-risk reward is defined as

$$R_\Delta = \sum_{k=1}^{H} \left[ C_{\text{risk}}\left(\tau_{\text{baseline}}^{(k)}\right) - C_{\text{risk}}\left(\tau_{\text{action}}^{(k)}\right) \right].$$

This formulation rewards actions that reduce predicted short-term clinical risk relative to inaction, while avoiding penalization for glucose excursions that are unavoidable under the underlying dynamics.

The proxy model is not intended to reproduce the full non-linear ODE dynamics. Instead, it serves as a stable and differentiable approximation used exclusively for reward shaping during training. The full physiologic simulator remains the authoritative environment for state transitions and safety evaluation. The proxy model is never used at deployment time and does not provide safety guarantees. The resulting raw reward is computed as

$$\begin{aligned} R_{\text{raw}} = &\, R_\Delta + R_{\text{survival}} - R_{\text{friction}} \\ &- R_{\text{progressive}} - R_{\text{spacing}} - R_{\text{inaction}} - R_{\text{structural}}, \end{aligned}$$

with each component defined in the following section.

**Reward Components.**

- **Survival bonus**: Encourage stable glycemia without unnecessary actions.

$$R_{\text{survival}} = \begin{cases} 0.2 & 90 \leq BG \leq 140, \\ 0.1 & 70 \leq BG \leq 180, \text{ only if } \mathbb{I}_{\text{inact}} = 1. \\ 0 & \text{otherwise}, \end{cases}$$

  Here $\mathbb{I}_{\text{inact}} = 1$ if and only if no bolus, meal, or exercise is taken.

- **Friction penalties**: Discourage unnecessary interventions and excessive insulin.

$$\begin{aligned} R_{\text{friction}} = &\, 0.005\, u_{\text{bolus}} + 0.001\, m_g + 0.005\, \mathbb{I}_{\text{bolus}} \\ &+ 0.005\, \mathbb{I}_{\text{meal}} + 0.03(\text{IOB} - 3.5)\, \mathbb{I}_{(\text{bolus} \wedge \text{IOB} > 3.5)}, \end{aligned}$$

  where $\mathbb{I}_{\text{bolus}} = 1$ if and only if $u_{\text{bolus}} > 0$; $\mathbb{I}_{\text{meal}} = 1$ if and only if $m_g > 0$. IOB (insulin on board) is the total active insulin remaining (basal+bolus) from the decay kernel.

- **Progressive daily pacing**: Prevent early-day overuse by spreading actions across the day. Let $N_b, N_m$ be *post-action* daily counts (bolus/meal), and $\tau = \text{frac\_day} = \frac{\text{minutes\_into\_day}}{1440} \in [0, 1]$.

$$T_b(\tau) = 5\tau + 1, \quad T_m(\tau) = 3\tau + 1,$$

$$R_{\text{progressive}} = 0.001 \left[ \max(0, N_b - T_b)^2 + \max(0, N_m - T_m)^2 \right].$$

- **Bolus spacing penalty**: Avoid rapid repeat boluses that can compound hypoglycemia risk.

$$R_{\text{spacing}} = 0.01 \cdot \mathbb{I}_{(\text{bolus} \wedge \Delta t_{\text{bolus}} < 30 \text{ min})}.$$

- **High-inaction penalty**: Encourage taking actions when glucose is clearly high.

$$R_{\text{inaction}} = \begin{cases} 0.005\,(BG - 180) & \text{if } \mathbb{I}_{\text{inact}} = 1 \text{ and } BG > 180, \\ 0 & \text{otherwise}. \end{cases}$$

- **Structural daily-limit penalty**: Enforce hard daily caps to prevent excessive interventions.

$$R_{\text{structural}} = 0.1 \left[ \max(0, N_b - \bar{N}_b)^2 + \max(0, N_m - \bar{N}_m)^2 \right],$$

  where $\bar{N}_b = 8$ and $\bar{N}_m = 7$ are the hard daily caps.

**Episode Termination.** Episodes terminate early when:

1. $BG < 10 \text{ mg/dL}$ (critical hypoglycemia),

2. $BG > 600 \text{ mg/dL}$ (critical hyperglycemia),

3. the simulation time limit is reached.

A terminal penalty scales with the remaining horizon to make early termination strictly worse:

$$P_{\text{term}} = 2.0 \cdot H_{\text{rem}},$$

where $H_{\text{rem}}$ is the number of steps remaining in the episode.

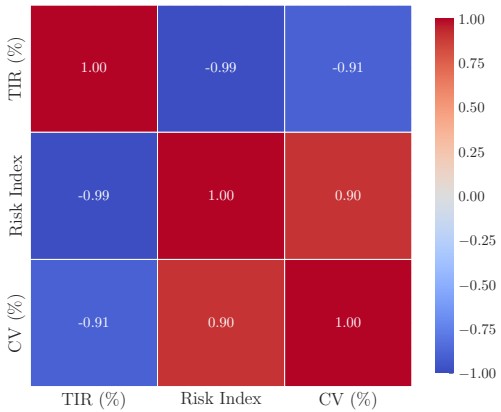

*Figure 4.* Correlation heatmap (Pearson) among key clinical metrics, including time-in-range (TIR), glucose risk index, and coefficient of variation (CV). The heatmap illustrates how reward-aligned metrics co-vary with measures of glycemic risk and variability.

### A.6. Reward and Cost Validation via Clinical Metrics

To assess whether the learned reward and cost signals align with standard clinical outcome measures, we analyze their relationship with established glycemic metrics, including time-in-range (TIR), glucose risk index, and coefficient of variation (CV).

Figure 5 illustrates the relationship between episode-averaged reward and TIR. Each point corresponds to a single training seed, while emphasized markers denote per-algorithm means. The clear positive trend indicates that higher accumulated reward consistently corresponds to improved time-in-range across algorithms and random seeds, confirming that the reward function promotes clinically desirable glycemic control rather than exploiting simulator-specific artifacts. Because we report in-distribution performance over an extended episode length (7 days), the reward and cost values differ from those reported in Figure 8, which shows training dynamics for a 1-day episode length.

Figure 6 shows the relationship between episode cost and risk index. The observed negative correlation demonstrates that lower cost values are associated with lower glycemic risk, validating that the cost signal captures clinically meaningful safety.

Finally, Figure 4 summarizes the pairwise correlations among TIR, risk index, and CV. Across the evaluated policies, TIR exhibits strong negative correlation with glycemic risk and variability, indicating that within the learned policy regime induced by our reward and cost design, higher TIR is typically accompanied by safer and more stable glucose trajectories. We therefore report multiple complementary metrics rather than relying on TIR alone as a proxy for control quality.

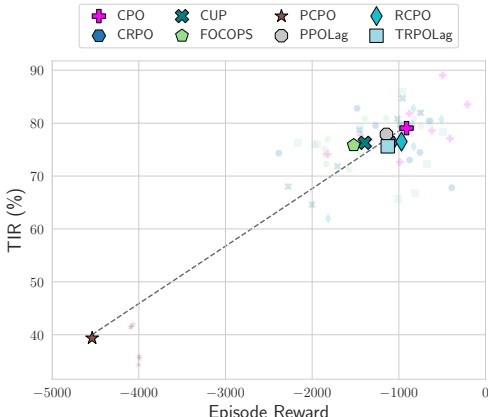

*Figure 5.* Relationship between episode reward and time-in-range (TIR). Each point represents one training seed; bold markers denote per-algorithm means. The positive correlation indicates that higher reward corresponds to improved TIR across algorithms and random seeds.

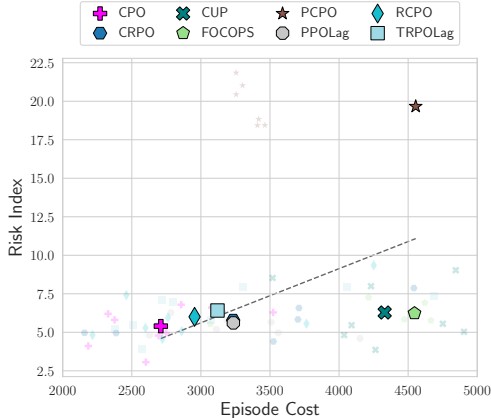

*Figure 6.* Relationship between episode cost and risk index. Lower cost values consistently correspond to lower glycemic risk, confirming that the cost signal captures clinically meaningful safety.

## B. Proof of Theorem 5.2

*Proof.* Fix a decision time $t$ and consider any candidate action $a$ evaluated by the shield at state $s_t$. Let $BG_\tau(a)$ denote the *true* future blood glucose at time $\tau \in [t, t+H]$ under action $a$, and let $\widehat{BG}_\tau(a)$ denote the corresponding model prediction.

Define the one-sided (hypoglycemia-relevant) uniform prediction error over the horizon,

$$\Delta_H(a) := \max_{\tau \in [t, t+H]} \left( \widehat{BG}_\tau(a) - BG_\tau(a) \right). \quad (4)$$

By Definition 5.1, $(\varepsilon, \alpha)$-reliability implies

$$\Pr(\Delta_H(a) \le \varepsilon) \ge 1 - \alpha. \quad (5)$$

Let $\mathcal{E}$ denote the event $\{\Delta_H(a) \le \varepsilon\}$. On $\mathcal{E}$, by definition

of $\Delta_H(a)$, for all $\tau \in [t, t+H]$,

$$\widehat{BG}_\tau(a) - BG_\tau(a) \le \varepsilon \Rightarrow BG_\tau(a) \ge \widehat{BG}_\tau(a) - \varepsilon. \quad (6)$$

Now suppose the shield permits action $a$. By the pruning rule, this means

$$m(a) = \min_{\tau \in [t,t+H]} \widehat{BG}_\tau(a) \ge G^\downarrow_{\text{shield}}. \quad (7)$$

Combining (6) and (7), on the event $\mathcal{E}$ we obtain

$$\min_{\tau \in [t,t+H]} BG_\tau(a) \ge \min_{\tau \in [t,t+H]} \left( \widehat{BG}_\tau(a) - \varepsilon \right)$$
$$= \min_{\tau \in [t,t+H]} \widehat{BG}_\tau(a) - \varepsilon = m(a) - \varepsilon \ge G^\downarrow_{\text{shield}} - \varepsilon. \quad (8)$$

By the theorem assumption $G^\downarrow_{\text{shield}} = G^\downarrow_{\text{fail}} + \varepsilon$, it follows that

$$\min_{\tau \in [t,t+H]} BG_\tau(a) \ge G^\downarrow_{\text{fail}} \text{ on } \mathcal{E}. \quad (9)$$

Therefore,

$$\Pr\left( \min_{\tau \in [t,t+H]} BG_\tau(a) \ge G^\downarrow_{\text{fail}} \right) \ge \Pr(\mathcal{E}) \ge 1 - \alpha, \quad (10)$$

which concludes the proof. $\square$

The guarantee in Theorem 5.2 is *conditional* on the $(\varepsilon, \alpha)$-reliability of the learned dynamics model over the finite horizon $H$, and it applies only to the finite candidate action set screened by the shield at each step (top-$k$ bolus levels paired with discrete meal levels). In practice we adopt pruning thresholds around $2 \times \text{error}_{dynamics}$ (e.g., $B_{\text{shield}} = 80$ mg/dL for a clinical limit $B_{\text{fail}} = 70$ mg/dL) where $\text{error}_{dynamics}$ is BA-NODE RMSE error 2. This can be interpreted as a robustness margin against bounded prediction error.

**Extension to hyperglycemia prevention.** An analogous construction applies to hyperglycemia events. Let the shield enforce a conservative upper pruning threshold $G^\uparrow_{\text{shield}} := G^\uparrow_{\text{fail}} - \varepsilon$. If the predictor satisfies a one-sided $(\varepsilon, \alpha)$-reliability condition for hyperglycemia,

$$\Pr\left( \max_{\tau \in [t,t+H]} \left( BG_\tau(a) - \widehat{BG}_\tau(a) \right) \le \varepsilon \right) \ge 1 - \alpha,$$

and the shield permits a candidate action $a$ only when $M(a) \le G^\uparrow_{\text{shield}}$, then any permitted action satisfies

$$\Pr\left( \max_{\tau \in [t,t+H]} BG_\tau(a) \le G^\uparrow_{\text{fail}} \right) \ge 1 - \alpha.$$

. The proof for the hyperglycemia prevention guarantee follows the same logic.

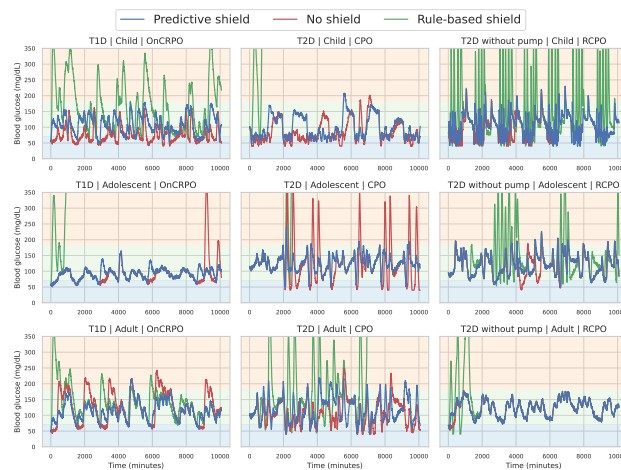

*Figure 7.* **Representative glucose trajectories.** Each row corresponds to a patient cohort (Child, Adolescent, Adult), and each column corresponds to a diabetes type (T1D, T2D, T2D without pump). For each setting, we compare trajectories generated by the base policy (No Shield), a static Rule-Based Shield, and the proposed Predictive Shield. Trajectories are shown for representative patients and algorithms. Across all settings, predictive shielding reduces unsafe excursions and suppresses oscillatory behavior induced by static rules and base policies.

## C. Predictive Shield Specification for Diabetes Control

We specify the implemented shield as a deterministic action-filtering layer that combines static clinical rules with a model-based predictive check. The shield operates on the policy distribution at each decision step and overrides the distribution based on safety filtering through predictive safety check.

### C.1. Shield Inputs and Action Space

Let the discrete action be $a = (a_{\text{bolus}}, a_{\text{meal}})$, where $a_{\text{bolus}} \in \{0, \dots, B-1\}$ and $a_{\text{meal}} \in \{0, \dots, M-1\}$ index the bolus and meal levels. Let the current observation include blood glucose $BG_t$ (mg/dL).

### C.2. Predictive Lower-Bound Check

For a candidate action $a$, the dynamics model predicts a blood glucose trajectory $\widehat{BG}_{t:t+H}(a)$ over a finite horizon $H$. We summarize the predicted trajectory by its extremal values:

$$m(a) = \min_{\tau \in [t,t+H]} \widehat{BG}_\tau(a), \; M(a) = \max_{\tau \in [t,t+H]} \widehat{BG}_\tau(a).$$

A candidate action is flagged unsafe for hypoglycemia if

$$m(a) < G^\downarrow_{\text{shield}},$$

where $G^\downarrow_{\text{shield}}$ is a conservative hypoglycemia prevention

threshold. Similarly, a candidate action is flagged unsafe for hyperglycemia if

$$M(a) > G^{\uparrow}_{\text{shield}},$$

where $G^{\uparrow}_{\text{shield}}$ is a conservative hyperglycemia prevention threshold.

The model-based predictive check is applied only when sufficient history is available to initialize the dynamics model; otherwise, the shield defaults to the static safety rules described below.

### C.3. Safety Rules (Implemented Penalties)

The shield applies the following logit penalties:

1. **Critical Hypoglycemia Rescue.** If $BG_t < 60$ mg/dL, the shield boosts the rescue meal level (fixed index) and penalizes all bolus levels above zero.

2. **Safe-Range Bolus Cap.** If $70 \leq BG_t \leq 180$ mg/dL, the shield penalizes bolus levels above the first two discrete choices, discouraging aggressive corrections while still permitting mild interventions.

3. **Predictive Hypo- and Hyperglycemia Prevention.** For the candidate set formed by the top-$k$ bolus actions from the policy paired with all discrete meal levels, the shield evaluates the predicted glucose trajectory $\widehat{BG}_{t:t+H}(a)$ for each candidate action $a$. The action is penalized if its predicted worst-case values violate either safety margin:

$$m(a) < G^{\downarrow}_{\text{shield}} \text{ or } M(a) > G^{\uparrow}_{\text{shield}},$$

where $m(a)$ and $M(a)$ denote the minimum and maximum predicted glucose over the horizon, respectively.

For minimal intervention, if no candidate action is flagged by the predictive check, the shield leaves the policy distribution unchanged.

**Logit Masking.** For shielding, we implement finite penalties as a constant negative logit bias $\beta \in \{-5, -10, -15\}$ applied to unsafe actions, i.e., $M(s, a) = \beta$. For each algorithm, $\beta$ is selected based on performance on a validation split and then fixed for all reported evaluations.

## D. Rule-Based Shield Implementation Details

To evaluate the necessity of predictive safety mechanisms, we implemented a robust Rule-Based Shield (RBS) baseline. This shield mimics safety features found in modern commercial insulin pumps, combining *Threshold Suspend (Low Glucose Suspend)* logic (Bergenstal et al., 2013) with

*insulin-on-board (IOB)* constraints inspired by clinical bolus calculator guidelines (Walsh et al., 2011).

Unlike the predictive shield, which forecasts future risk, RBS operates on static, reactive thresholds derived from current clinical guidelines. The shield intervenes in the policy's action space A by applying hard masks to the action logits based on the current glucose state $BG_t$, trend $\Delta BG_t$, and active insulin $IOB_t$.

The RBS applies the following rules:

### D.1. Emergency Hypoglycemia Rescue

If blood glucose falls below the critical hypoglycemia threshold ($BG_t < 70$ mg/dL), immediate rescue action is triggered. So, RBS forces the intake of rescue meal (e.g., 15g) and completely blocks any insulin bolus. A large penalty ($-\infty$) is applied to all non-rescue actions in the logit space, forcing the selection of the rescue action.

### D.2. High Glucose Correction with IOB Gating

If blood glucose exceeds the hyperglycemia threshold ($BG_t > 250$ mg/dL), a correction bolus is necessary. However, to prevent insulin stacking, which is a common issue of hypoglycemia, the shield performs an active insulin check. Hence, the correction is allowed if and only if $IOB_t < 2.0$ U. If safe, the shield masks the 'No Bolus' action, forcing the agent to administer a correction dose. This gating mechanism implements a conservative variant of the IOB-aware dosing logic found in clinical bolus calculators (Walsh et al., 2011).

### D.3. Low Glucose Suspend (LGS)

If the patient is not in critical hypoglycemia but is approaching it, the shield activates the *Threshold Suspend* mechanism. When ($BG_t < 100$ mg/dL $\wedge \Delta BG_t < 0$), all bolus actions are masked, forcing the agent to suspend insulin delivery.

---
**Algorithm 1** Rule-Based Safety Shield (RBS)

---
**Require:** State $s_t = (BG_t, \text{IOB}_t, \Delta BG_t, \ldots)$, logits $L$

1: ▷ Clinical thresholds: $BG_{\text{res}} = 70$ (rescue), $BG_{\text{susp}} = 90$ (suspend), $BG_{\text{hyper}} = 250$ (hyper), $BG_{\text{warn}} = 110$ (warn), $\text{IOB}_{\text{safe}} = 2.0$, $C_{\text{block/force}}$ large penalties

2: **if** $BG_t < 70$ **then**                    ▷ Critical hypo rescue

3:      $L[\text{NoCarb}] \leftarrow -\infty$; $L[\text{Bolus}] \leftarrow -\infty$

4: **else if** $BG_t > 250$ and $\text{IOB}_t < 2.0$ **then**        ▷ Severe hyper correction

5:      $L[\text{NoBolus}] \leftarrow -C_{\text{force}}$

6: **else if** $(BG_t < 100 \wedge \Delta BG_t < 0)$ **then**                    ▷ Low-glucose suspend

7:      $L[\text{Bolus}] \leftarrow -C_{\text{block}}$

8: **end if**

9: **return** $\text{Softmax}(L)$

---

| Parameter | PPOLag | TRPOLag | CPO | RCPO | FOCOPS | CRPO | PCPO | CUP |
|---|---|---|---|---|---|---|---|---|
| Update Iters | 40 | 10 | 10 | 10 | 40 | 10 | 10 | 40 |
| Clip Ratio | 0.2 | n/a | n/a | n/a | 0.2 | n/a | n/a | 0.2 |
| Actor LR (t1d) | 3e-4 | 3e-4 | 3e-5 | 3e-5 | 3e-4 | 3e-5 | 3e-5 | 3e-4 |
| Actor LR (t2d) | 3e-4 | 3e-4 | 3e-5 | 3e-5 | 3e-4 | 3e-4 | 3e-5 | 3e-4 |
| Actor LR (t2d w/o pump) | 3e-4 | 3e-4 | 3e-4 | 3e-4 | 3e-4 | 3e-5 | 3e-5 | 3e-5 |
| Critic LR (t1d) | 1e-4 | 5e-4 | 1e-4 | 1e-4 | 1e-4 | 5e-4 | 5e-4 | 1e-4 |
| Critic LR (t2d) | 1e-4 | 1e-4 | 5e-5 | 5e-4 | 1e-4 | 1e-4 | 5e-4 | 5e-5 |
| Critic LR (t2d w/o pump) | 1e-4 | 1e-4 | 5e-4 | 1e-4 | 5e-4 | 5e-4 | 5e-4 | 5e-5 |
| Lambda LR | 0.035 | 0.035 | n/a | 0.035 | 0.035 | n/a | n/a | 0.035 |
| CG Iters | n/a | 15 | 15 | 15 | n/a | 15 | 15 | n/a |
| CG Damping | n/a | 0.1 | 0.1 | 0.1 | n/a | 0.1 | 0.1 | n/a |

*Table 6.* Default OmniSafe (Ji et al., 2024) hyperparameters for on-policy baselines with tuned actor/critic learning rates.

## E. Training Details

All policies are trained for 5 million environment steps. Each episode corresponds to a single simulated day (24 hours), after which the environment is reset. Training is performed on a *single representative patient instance* per cohort: `Child#01`, `Adolescent#01`, and `Adult#01`. This setup reflects a realistic deployment scenario in which a controller is trained on limited patient data and must generalize to unseen physiological parameters. All reported results are averaged over three random seeds. Figure 8 shows the training trajectories for both reward and cost.

We use consistent training configurations across all baselines, with `steps_per_epoch`=2048, `batch_size`=64, `target_kl`=0.1, `entropy_coef`=0.01, and a `cost_limit`=100.

All methods employ two-layer MLP actors with hidden sizes $[64, 64]$. Critic architectures follow OmniSafe (Ji et al., 2024) defaults: $[128, 128]$ for PPOLag, TRPOLag, CPO, and CUP, and $[64, 64]$ for RCPO, FOCOPS, OnCRPO, and PCPO.

To manage computational constraints while ensuring rigorous evaluation, we focused our hyperparameter tuning exclusively on the learning rate, which large-scale empirical studies have identified as one of the most important factors to affect performance in on-policy algorithms (Andrychowicz et al., 2021). Actor and critic learning rates are selected via a small grid search, with actor learning rates in $\{3 \times 10^{-4}, 3 \times 10^{-5}\}$ and critic learning rates in $\{1 \times 10^{-4}, 5 \times 10^{-4}, 5 \times 10^{-5}\}$.

## F. Additional Dynamics Model Analysis

In this section, we provide a detailed ablation study of the Basis-Adaptive Neural ODE (BA-NODE) architecture, analyzing its sensitivity to hyperparameters and its generalization capabilities compared to baselines. Figure 9 presents

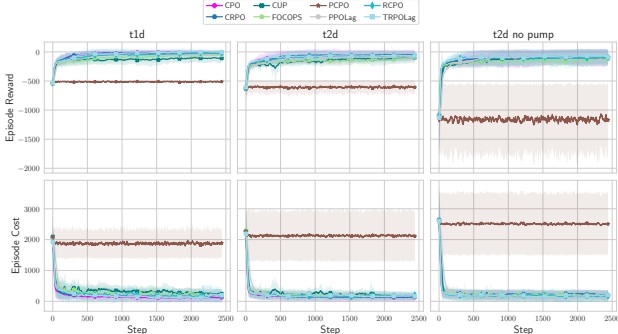

*Figure 8.* Training dynamics of episode reward and cost during training. We report mean trajectories with $\pm$ std bands aggregated across all runs (seeds and cohorts) for each algorithm. Columns correspond to diabetes types (t1d, t2d, t2d without pump), and rows correspond to episode reward (top) and episode cost (bottom).

the aggregated blood-glucose prediction trajectories for each cohort–diabetes-type pair, showing how BA-NODE (blue), ITransformer (green), and Neural ODE (red) track the ground-truth dynamics. Figure 10 shows BA-NODE architecture, consisting of three modules: an ITransformer history encoder, an ensemble latent Neural ODE, and a Function-Encoder that computes context-conditioned basis coefficients.

**Dynamics Predictor Inputs and Targets.** The dynamics predictor is trained on transition tuples generated from simulator rollouts and a pretrained policy by CPO algorithm. At each time step $t$, the model input consists of a stacked history of recent observations and the executed action. Specifically, each observation vector includes:

> {`cgm`, `iob`, `cob`, `cgm_trend`, `time_since_meal`,
> `time_until_meal_norm`, `next_meal_size_norm`,
> `is_pre_bolus_window`}

together with the action applied at the corresponding time step.

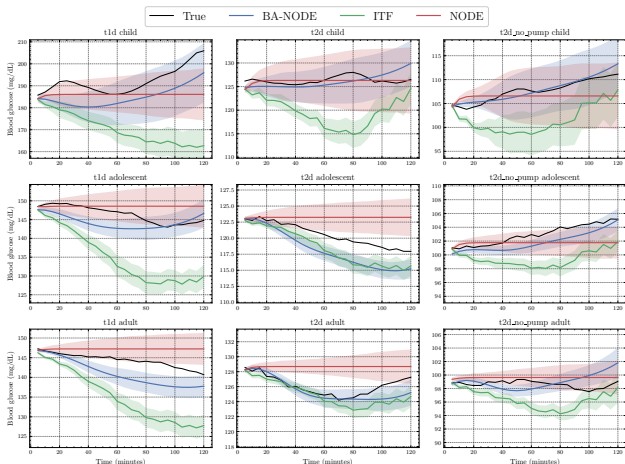

*Figure 9.* **Cohort/Type Prediction Trajectories.** Each cell aggregates the ground-truth blood glucose trajectory, labeled as "True", and model predictions for a cohort– diabetes-type pair, using 10 patients and 10 randomly sampled windows per patient (100 samples total). Curves show mean blood glucose trajectories over a 24-step horizon (120 minutes); shaded bands indicate the standard error of prediction error.

Given a history window of length $H$, the input to the model is denoted by

$$x_t = \{(o_{t-H+1}, a_{t-H+1}), \ldots, (o_t, a_t)\}.$$

The prediction target is the future blood glucose trajectory over a horizon $P$. During training, this is implemented via incremental glucose changes,

$$y_{t+k} = \Delta\mathrm{BG}_{t+k} = \mathrm{BG}_{t+k+1} - \mathrm{BG}_{t+k}, \quad k = 0, \ldots, P-1,$$

which are recursively integrated to obtain absolute glucose predictions. This representation improves numerical stability and facilitates long-horizon rollout.

**Parameters.** To ensure a fair comparison, we standardized the model capacity across all baselines by restricting the parameter budget. We adjusted the hidden dimensions and number of layers such that all models are designed to have a comparable trainable parameters. Specifically, our proposed BA-NODE (389,500 params) is compared against iTransformer (533,894 params) and Neural ODE (402,061 params). Our method achieves the reported performance despite having slightly fewer parameters than the competing baselines.

### F.1. Evaluation Metrics

We report results on a held-out evaluation dataset including 5 days of simulated glucose trajectories for each patient cohort (adult, adolescent, pediatric), distinct from the 15-day training dataset generated by the unified clinical simulator. All errors are calculated on the denormalized (absolute)

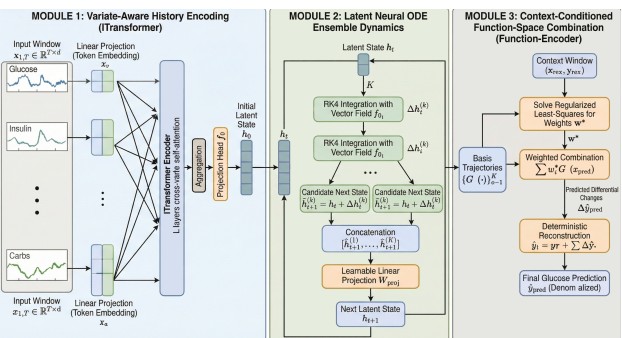

*Figure 10.* **Model Architecture.** BA-NODE consists of three modules: (1) an ITransformer encoder that maps multivariate history into a latent state, (2) an ensemble of latent Neural ODE basis flows integrated forward to produce candidate trajectories, and (3) a Function-Encoder that solves a regularized least-squares problem to compute context-conditioned basis coefficients for personalized prediction.

blood glucose values. Let $y_{t+k}^{(i)}$ be the ground truth blood glucose value for the $i$-th sample at the $k$-th time step within the prediction horizon $P$, and let $\hat{y}_{t+k}^{(i)}$ be the corresponding predicted value. The total number of evaluation samples is $N$. We employ the following metrics:

**Mean Squared Error (MSE)**    The MSE is computed as the average of the squared differences over all time steps in the prediction horizon across all samples:

$$\mathrm{MSE} = \frac{1}{N} \sum_{i=1}^{N} \left( \frac{1}{P} \sum_{k=1}^{P} \left( y_{t+k}^{(i)} - \hat{y}_{t+k}^{(i)} \right)^2 \right) \quad (11)$$

**Root Mean Squared Error (RMSE)**    The RMSE is the square root of the MSE, providing an error metric in the same units as the input data:

$$\mathrm{RMSE} = \sqrt{\mathrm{MSE}} = \sqrt{\frac{1}{N \cdot P} \sum_{i=1}^{N} \sum_{k=1}^{P} \left( y_{t+k}^{(i)} - \hat{y}_{t+k}^{(i)} \right)^2} \quad (12)$$

**Final Displacement Error (FDE)**    Since the blood glucose trajectory is 1-dimensional, we define the FDE as the Mean Absolute Error (L1 distance) at the final prediction step ($k = P$). This captures the model's accuracy at the furthest forecast point:

$$\mathrm{FDE} = \frac{1}{N} \sum_{i=1}^{N} \left| y_{t+P}^{(i)} - \hat{y}_{t+P}^{(i)} \right| \quad (13)$$

### F.2. Basis Numbers Ablation

We evaluate the sensitivity of the BA-NODE architecture to the number of basis functions. As illustrated in Figure 11,

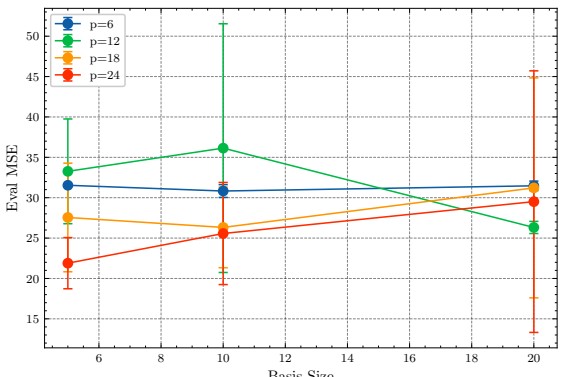

*Figure 11.* **Sensitivity to Basis Numbers.** The reconstruction error remains consistent across varying numbers of basis functions. This indicates that BA-NODE is robust to the choice of basis size, maintaining stable performance across varying basis numbers.

the model shows robustness to this hyperparameter. We observe that increasing the number of basis does not achieve significant performance gains, and in some cases, slightly increases error likely due to added model complexity. Moreover, the performance variance remains minimal across varying basis numbers, demonstrating that BA-NODE effectively captures the underlying dynamics without relying on extensive tuning of the basis numbers.

### F.3. Prediction Horizon Sensitivity

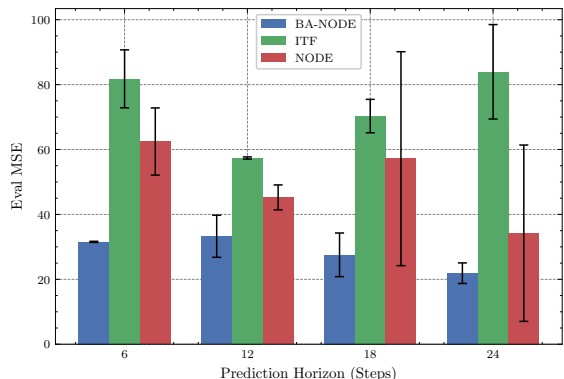

*Figure 12.* **Error over Extended Horizons.** Mean squared error (MSE) on evaluation dataset as the prediction horizon increases from $P = 6$ to $P = 24$ steps. BA-NODE demonstrate lowest error across varying prediction steps and reduced variance compared to baseline models, indicating improved long-horizon stability.

We evaluate the robustness of the learned dynamics models by measuring prediction error as the forecast horizon increases from $P = 6$ to $P = 24$ steps. As shown in Figure 12, BA-NODE consistently achieves lower mean squared error and smaller variance across all horizons. This suggests that the proposed basis-adaptive latent dynamics mitigate error accumulation over long rollouts, yielding more stable long-term predictions than baseline approaches.

*Table 7.* **Hidden Size Sensitivity.** BA-NODE dynamics prediction error under different hidden sizes. The main experiments use hidden size 128.

### F.4. Hidden Size Sensitivity

| Hidden Size | MAE ↓ | FDE ↓ | RMSE ↓ |
|---|---|---|---|
| 16 | $2.87 \pm 0.02$ | $3.72 \pm 0.04$ | $4.56 \pm 0.08$ |
| 32 | $2.87 \pm 0.06$ | $3.69 \pm 0.09$ | $4.58 \pm 0.23$ |
| 64 | $2.85 \pm 0.02$ | $3.69 \pm 0.04$ | $4.49 \pm 0.04$ |
| 128 | $\mathbf{2.82 \pm 0.01}$ | $\mathbf{3.63 \pm 0.03}$ | $\mathbf{4.42 \pm 0.02}$ |

We evaluate the sensitivity of BA-NODE to the hidden size of the latent dynamics model. As shown in Table 7, performance is relatively stable across hidden sizes, with only moderate variation in MAE, FDE, and RMSE. Increasing the hidden size generally improves prediction accuracy, and hidden size 128 achieves the lowest error across all three metrics. Therefore, we use hidden size 128 in the main experiments. The small gap between configurations also suggests that BA-NODE is not overly sensitive to this hyperparameter, while the larger hidden size provides the best overall prediction quality.

### F.5. Evaluation Error Dynamics

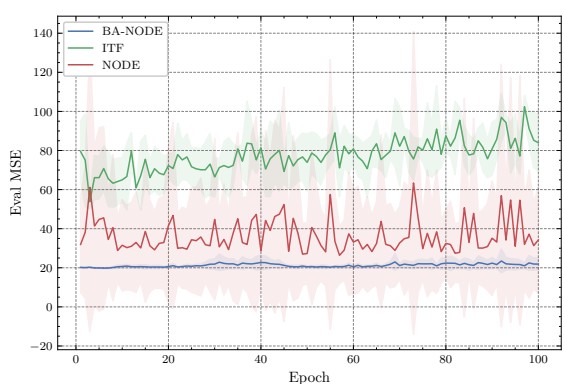

*Figure 13.* **Training Stability and Generalization.** Progress on evaluation MSE during training. BA-NODE shows smooth convergence, whereas baselines show high variance and instability.

To assess generalization stability, we plot the evaluation MSE throughout the training process (Figure 13). BA-NODE demonstrates stable convergence in early stage of training. Conversely, baseline models show noisy evaluation dynamics and a tendency to overfit the training distribution as training epochs progress.

### F.6. Coefficient Consistency Analysis

To assess whether basis coefficients reflect patient-specific conditioning without requiring discrete clustering, we ana-

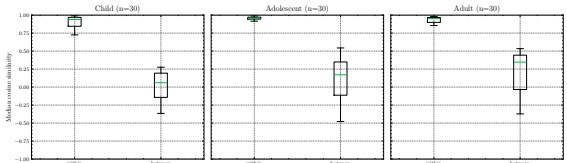

*Figure 14.* Within- vs. between-patient median cosine similarities organized by cohort. Within-patient medians are consistently higher, indicating personalization holds across cohorts.

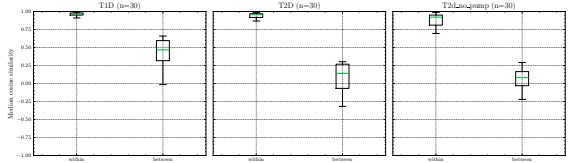

*Figure 15.* Within- vs. between-patient median cosine similarities organized by diabetes type. The within-patient medians exceed between-patient medians across all types, suggesting consistent personalization across diabetes categories.

lyze cosine similarity in the coefficient space. Figures 15 and 14 visualize this analysis across diabetes types and age cohorts, respectively.

For each patient $i$, we compute a mean coefficient vector $\mu_i$ by averaging coefficients across multiple context windows. Given an individual coefficient vector $w_{i,t}$, we measure within-patient similarity as

$$\text{sim}_{\text{within}} = \cos(w_{i,t}, \mu_i),$$

and between-patient similarity by comparing to other patients' means,

$$\text{sim}_{\text{between}} = \cos(w_{i,t}, \mu_j), \quad j \neq i.$$

Across 90 patients with 20 coefficient samples each (1,800 vectors total), we observe high within-patient similarity (mean 0.8789; median 0.9487) and low between-patient similarity (mean 0.0667; median 0.0673). The nearest-mean assignment accuracy is 0.3244, far above random (1/90), indicating that coefficients are more aligned with their own patient's mean than with others while still overlapping across patients.

### F.7. Detailed Analysis of Adaptation Mechanism

To quantify the benefit of function-space adaptation, we compare two inference modes on the unseen test set, averaged over 3 random seeds:

1. **Global Baseline (Fixed $\bar{w}$):** We compute type-specific global coefficients by averaging coefficients across *test* patients within each diabetes type (T1/T2/T2 without pump). Each test patient uses the fixed $\bar{w}$ for their type, disabling individual adaptation.

| Type | Cohort | Static | Adapted | Gain |
|---|---|---|---|---|
| **Type 1** | Child | $31.12 \pm 1.05$ | $\mathbf{29.99 \pm 1.08}$ | +4% |
| | Adol. | $16.61 \pm 0.73$ | $\mathbf{16.25 \pm 0.31}$ | +2% |
| | Adult | $13.10 \pm 0.54$ | $\mathbf{12.84 \pm 0.44}$ | +2% |
| **Type 2** | Child | $20.68 \pm 0.21$ | $\mathbf{20.02 \pm 0.63}$ | +3% |
| | Adol. | $9.49 \pm 0.95$ | $\mathbf{8.83 \pm 0.95}$ | +7% |
| | Adult | $8.30 \pm 0.28$ | $\mathbf{8.06 \pm 0.32}$ | +3% |
| **Type 2 (w/o Pump)** | Child | $19.34 \pm 0.61$ | $\mathbf{19.10 \pm 0.72}$ | +1% |
| | Adol. | $8.99 \pm 0.55$ | $\mathbf{8.57 \pm 0.59}$ | +5% |
| | Adult | $9.35 \pm 0.05$ | $\mathbf{8.43 \pm 0.13}$ | +10% |

*Table 8.* **Adaptation Impact.** Comparison of prediction error (MAE) across patient cohorts. *Static* denotes fixed model weights; *Adapted* denotes personalized weights.

2. **Personalized (Adaptive $w^*$):** For each patient, we solve for coefficients using their context windows.

Table 8 reports MAE (mean $\pm$ std), organized by *Diabetes Type* and *Age Cohort*. Overall, adaptive coefficients show consistent improvement.

### F.8. Training Procedure for BA-NODE

We train BA-NODE on patient trajectories converted into supervised sequence-to-sequence prediction problems. Each trajectory is segmented into sliding windows consisting of a history context of length $T$ and a prediction horizon of length $P$. The input $x_{t-T+1:t}$ contains multivariate physiological signals (e.g., glucose, insulin, carbohydrates), while the target $y_{t:t+P-1}$ corresponds to future *changes* in blood glucose rather than absolute values. Predicting deltas improves numerical stability and mitigates long-horizon drift.

All inputs and targets are normalized using statistics computed from the training trajectories of each cohort. The model operates entirely in the normalized space, and predictions are mapped back to physical units only at the final output stage.

**Basis trajectory generation.** Given a history window, the encoder maps the input sequence to an initial latent state $h_0$. An ensemble of $K$ latent vector fields is then integrated forward using a fourth-order Runge–Kutta scheme, producing $K$ candidate latent evolutions. At each integration step, the candidate next states are combined through a shared linear projection, resulting in a single latent trajectory. Each basis trajectory is decoded into a predicted glucose-delta sequence over the horizon. These decoded trajectories serve as function-space bases $\{G_k(\cdot)\}_{k=1}^{K}$.

**Coefficient conditioning via function encoding.** Rather than learning fixed patient-specific parameters, BA-NODE

conditions predictions by inferring a coefficient vector $w \in \mathbb{R}^K$ that combines the learned basis dynamics. For each patient, we draw a set of context windows $\{(x_{\text{ctx}}^{(n)}, y_{\text{ctx}}^{(n)})\}_{n=1}^N$, where $x_{\text{ctx}}^{(n)} \in \mathbb{R}^{T \times d}$ and $y_{\text{ctx}}^{(n)} \in \mathbb{R}^P$ (delta glucose over the horizon). Given these windows, the model produces basis outputs $G(x_{\text{ctx}}^{(n)}) \in \mathbb{R}^{P \times K}$, which are stacked across $N$ contexts. Following Ingebrand et al. (2025), Function Encoder solves a regularized least-squares problem using its inner-product formulation:

$$w^* = \arg\min_w \|Gw - y_{\text{ctx}}\|_2^2 + \lambda \|w\|_2^2,$$

where $G$ and $y_{\text{ctx}}$ denote the stacked basis outputs and targets. This leads to a closed-form solution via the normal equations, and the operation is differentiable, allowing gradients to flow back to the encoder and ODE parameters. In practice, this inference is performed per batch, enabling context-conditioned adaptation without gradient updates at test time.

**Loss and optimization.** With coefficients fixed, the model predicts glucose deltas on query windows and minimizes mean squared error against the normalized targets:

$$\mathcal{L}_{\text{pred}} = \mathbb{E}\big[\|\hat{y} - y\|_2^2\big].$$

We include a standard Function-Encoder regularization term that encourages the basis functions to remain well-conditioned by penalizing large off-diagonal entries in the Gram matrix. All neural components (encoder, latent dynamics, projection, and decoder) are optimized jointly using Adam with gradient clipping.

**Inference.** At test time, predicted glucose deltas are cumulatively summed to reconstruct absolute glucose trajectories. Denormalization is applied only at the final output, ensuring stable latent evolution throughout the prediction horizon.

# G. Detailed Clinical Metrics under Distribution Shift

This section provides a detailed breakdown of clinical safety and efficacy metrics under both *in-distribution* (ID) and *out-of-distribution* (OOD) evaluation settings. In addition to aggregate measures such as Time in Range (TIR), Risk Index, and Coefficient of Variation (CV), we explicitly report the frequency of hypoglycemic ($< 70$ mg/dL) and hyperglycemic ($> 180$ mg/dL) events.

### G.1. In-Distribution Performance on Training Patients

Table 1 compares unshielded baseline performance on the training patient (ID) against performance on unseen patients (OOD). Most constrained algorithms achieve strong ID performance, with high TIR ($> 80\%$) and low Risk Index

($< 5$), confirming that these methods can satisfy safety constraints when evaluated on the same physiology used during training. However, this competence does not translate to OOD settings, where all well-performing baselines show degradation in both TIR and Risk Index.

An exception is PCPO. However, it performs poorly even on the training patient (TIR $\approx 37\%$), indicating a training failure rather than a generalization problem.

Overall, these results show that training-time constraint satisfaction is *patient-specific* and does not guarantee safe deployment on new patients, motivating the need for a separate test-time safety mechanism such as predictive shielding.

### G.2. Out-of-Distribution Performance on Unseen Patients

We next evaluate the same policies on previously unseen patients (`Patients #02--#10`) to assess robustness under physiological distribution shift. This out-of-distribution analysis reveals how safety and stability degrade when patient-specific parameters such as insulin sensitivity and absorption dynamics differ from those observed during training, and highlights the extent to which predictive shielding mitigates these performance degrade.

**Interpreting Glucose Variability (CV) and Rescue Dynamics** While a lower Coefficient of Variation (CV) typically implies more stable control, we sometimes observe instances where the shielded agent shows a slightly higher CV compared to the baseline (For example, T2D without Pump, PPOLag or RCPO results), despite achieving better TIR and risk index.

This is likely due to a mechanism of the shield's active rescue behavior. When the patient's glucose trajectory approaches the hypoglycemic threshold, the shield enforces a corrective meal intervention. This action triggers a sudden rise in blood glucose levels while it helps blood glucose levels escape the danger zone. Although this sudden inflection increases the statistical variance (CV), it is a clinically necessary maneuver to prevent severe hypoglycemia. Consequently, a marginal rise in CV, when accompanied by a reduction in Risk Index and Hypo Events, reflects effective emergency safety enforcement rather than control instability.

**Dependence on Base Policy Quality** It is important to note that the shield operates as a *safety filter*, not a trajectory planner. It creates a safe subset of actions from the proposals generated by the base policy. If the original policy is fundamentally poorly trained (e.g., PCPO in Table 9), the shield's capacity to improve performance is bounded. Conversely, when the base policy is competent, the shield demonstrates significant synergy. By filtering out stochas-

tic safety violations, it allows the agent to maintain stable control, leading to substantial improvement in efficacy. As one example, in the CPO T2D cohort, shielding boosts TIR dramatically from 82.33% to 95.22%.

| Algorithm | TIR (%) | Risk Index | CV (%) | Hypo Events (%) | Hyper Events (%) |
|---|---|---|---|---|---|
| CPO | 81.73 ± 3.54 | 4.82 ± 0.92 | 27.60 ± 0.98 | 12.43 ± 2.06 | 0.34 ± 0.24 |
| CUP | 74.42 ± 6.83 | 6.95 ± 1.91 | 36.31 ± 4.92 | 11.33 ± 8.35 | 5.43 ± 2.10 |
| FOCOPS | 81.05 ± 2.02 | 5.20 ± 0.36 | 32.91 ± 4.19 | 7.80 ± 2.54 | 3.11 ± 0.63 |
| CRPO | 77.08 ± 7.16 | 5.83 ± 1.53 | 29.71 ± 2.62 | 13.13 ± 0.67 | 2.87 ± 3.50 |
| PCPO | 34.60 ± 1.13 | 21.26 ± 0.79 | 49.18 ± 0.39 | 7.62 ± 0.13 | 36.68 ± 2.18 |
| PPOLag | 78.72 ± 1.98 | 5.49 ± 0.53 | 31.77 ± 2.65 | 13.33 ± 3.48 | 1.99 ± 0.36 |
| RCPO | 80.60 ± 3.65 | 5.35 ± 1.46 | 29.71 ± 5.06 | 11.91 ± 1.50 | 2.05 ± 3.45 |
| TRPOLag | 78.15 ± 3.63 | 5.74 ± 1.04 | 31.21 ± 2.06 | 13.39 ± 1.90 | 2.14 ± 3.03 |

*Table 9.* Aggregated performance across cohorts (T1D, No Shield)

| Algorithm | TIR (%) | Risk Index | CV (%) | Hypo Events (%) | Hyper Events (%) |
|---|---|---|---|---|---|
| CPO | 82.73 ± 3.09 | 6.00 ± 0.95 | 36.42 ± 0.86 | 1.83 ± 0.74 | 7.11 ± 1.24 |
| CUP | 69.09 ± 6.37 | 10.78 ± 2.52 | 43.26 ± 6.11 | 3.47 ± 1.47 | 16.70 ± 4.85 |
| FOCOPS | 73.49 ± 1.35 | 8.58 ± 0.87 | 42.76 ± 1.82 | 3.07 ± 1.18 | 11.94 ± 1.81 |
| CRPO | 74.96 ± 8.45 | 8.55 ± 2.70 | 37.86 ± 2.42 | 2.33 ± 0.53 | 12.66 ± 5.51 |
| PCPO | 34.26 ± 0.60 | 21.43 ± 0.13 | 45.08 ± 0.40 | 4.03 ± 0.19 | 39.08 ± 0.12 |
| PPOLag | 73.15 ± 1.79 | 9.35 ± 0.98 | 39.79 ± 2.51 | 3.47 ± 0.41 | 14.17 ± 2.17 |
| RCPO | 78.00 ± 3.98 | 7.54 ± 1.23 | 38.93 ± 1.90 | 2.89 ± 0.46 | 10.23 ± 2.43 |
| TRPOLag | 78.29 ± 7.52 | 7.58 ± 2.40 | 38.59 ± 2.90 | 1.96 ± 0.08 | 10.95 ± 5.13 |

*Table 10.* Aggregated performance across cohorts (T1D, Rule-based Shield)

| Algorithm | TIR (%) | Risk Index | CV (%) | Hypo Events (%) | Hyper Events (%) |
|---|---|---|---|---|---|
| CPO | 86.92 ± 2.61 | 3.23 ± 0.56 | 24.20 ± 1.74 | 5.52 ± 2.81 | 0.17 ± 0.30 |
| CUP | 74.87 ± 6.81 | 6.84 ± 1.87 | 35.93 ± 4.99 | 10.24 ± 8.06 | 5.70 ± 2.19 |
| FOCOPS | 84.03 ± 3.32 | 4.26 ± 0.68 | 28.45 ± 5.27 | 6.00 ± 4.95 | 1.62 ± 0.98 |
| CRPO | 85.17 ± 4.52 | 3.64 ± 0.94 | 27.42 ± 1.94 | 10.30 ± 5.28 | 0.31 ± 0.24 |
| PCPO | 34.86 ± 0.48 | 21.04 ± 0.17 | 47.55 ± 0.43 | 6.18 ± 0.35 | 36.89 ± 0.58 |
| PPOLag | 85.81 ± 1.43 | 3.90 ± 0.32 | 28.69 ± 0.20 | 4.04 ± 1.02 | 2.39 ± 0.72 |
| RCPO | 87.80 ± 0.89 | 3.09 ± 0.06 | 25.00 ± 2.09 | 8.51 ± 1.14 | 0.11 ± 0.19 |
| TRPOLag | 83.76 ± 3.23 | 4.45 ± 1.02 | 30.69 ± 1.00 | 6.14 ± 1.60 | 2.67 ± 2.68 |

*Table 11.* Aggregated performance across cohorts (T1D, Predictive Shield)

### G.3. Type 1 Diabetes (T1D)

Tables 9, 10, and 11 summarize the results for the T1D cohort.

**Failure of Reactive Heuristics.** The Rule-Based Shield (RBS) highlights the danger of static, reactive safety constraints. While it successfully mitigates hypoglycemia, reducing CPO hypo events from 12.43% to 1.83%, this comes at the cost of severe rebound hyperglycemia. By blindly blocking insulin based on current thresholds, RBS causes PPO-Lag's hyperglycemic events to spike from 1.99% to 14.17%. Consequently, the overall clinical outcome degrades: PPO-Lag's Risk Index nearly doubles (5.49 → 9.35) and TIR drops (78.72% → 73.15%), proving that heuristic interventions often trade one safety risk for another. This trade-off is consistent across all evaluated algorithms: RBS consistently trades reduced hypoglycemia for increased hyperglycemia, frequently resulting in a net degradation of Risk Index compared to the unshielded baseline.

| Algorithm | TIR (%) | Risk Index | CV (%) | Hypo Events (%) | Hyper Events (%) |
|---|---|---|---|---|---|
| CPO | 82.33 ± 6.23 | 4.68 ± 1.73 | 26.31 ± 4.50 | 8.77 ± 1.30 | 2.09 ± 2.81 |
| CUP | 77.14 ± 9.65 | 5.91 ± 2.45 | 33.62 ± 6.33 | 9.17 ± 1.74 | 4.36 ± 3.84 |
| FOCOPS | 74.53 ± 1.44 | 6.47 ± 0.62 | 33.67 ± 1.91 | 10.70 ± 1.09 | 3.88 ± 1.31 |
| CRPO | 79.25 ± 4.24 | 5.15 ± 1.15 | 28.37 ± 5.95 | 12.08 ± 4.51 | 0.90 ± 0.43 |
| PCPO | 41.48 ± 0.76 | 18.61 ± 0.32 | 51.02 ± 0.61 | 7.10 ± 0.49 | 30.67 ± 1.32 |
| PPOLag | 80.06 ± 2.94 | 4.85 ± 0.54 | 29.96 ± 1.20 | 6.19 ± 2.09 | 2.39 ± 1.09 |
| RCPO | 72.16 ± 8.68 | 6.79 ± 2.13 | 30.46 ± 5.34 | 11.49 ± 4.30 | 4.26 ± 2.83 |
| TRPOLag | 78.04 ± 9.56 | 5.70 ± 1.99 | 27.80 ± 3.30 | 11.97 ± 4.12 | 2.70 ± 3.55 |

*Table 12.* Aggregated performance across cohorts (T2D, No Shield)

| Algorithm | TIR (%) | Risk Index | CV (%) | Hypo Events (%) | Hyper Events (%) |
|---|---|---|---|---|---|
| CPO | 78.60 ± 6.47 | 6.66 ± 1.71 | 38.45 ± 4.73 | 3.45 ± 1.37 | 8.26 ± 2.99 |
| CUP | 73.10 ± 9.08 | 8.09 ± 1.99 | 43.36 ± 3.11 | 3.44 ± 0.61 | 11.13 ± 3.25 |
| FOCOPS | 64.75 ± 4.10 | 10.69 ± 1.13 | 49.27 ± 1.20 | 4.32 ± 0.73 | 15.65 ± 2.16 |
| CRPO | 75.21 ± 5.68 | 7.56 ± 1.99 | 42.66 ± 6.63 | 3.16 ± 0.94 | 10.09 ± 3.94 |
| PCPO | 40.95 ± 1.47 | 19.11 ± 0.28 | 49.61 ± 0.91 | 5.28 ± 0.34 | 32.38 ± 0.61 |
| PPOLag | 72.57 ± 3.34 | 8.23 ± 0.83 | 40.43 ± 1.44 | 3.73 ± 0.23 | 10.42 ± 1.02 |
| RCPO | 72.25 ± 7.61 | 7.97 ± 2.14 | 41.52 ± 5.53 | 3.13 ± 0.86 | 10.77 ± 4.09 |
| TRPOLag | 76.41 ± 8.86 | 7.40 ± 2.04 | 39.31 ± 4.30 | 3.20 ± 0.82 | 10.30 ± 3.66 |

*Table 13.* Aggregated performance across cohorts (T2D, Rule-based Shield)

**Predictive Shielding Advantage.** In contrast, the Predictive Shield uses BA-NODE dynamics model to forecast future glucose trajectories, preventing violations without over-correction. Shielded CPO achieves superior balance: it reduces hypo events (5.52%) while eliminating hyperglycemia (0.17%). This results in the highest overall TIR (86.92%) and lowest Risk Index (3.23), demonstrating that predictive foresight is required to improve safety without compromising therapeutic efficacy. Also, this improvement is consistent; unlike RBS, predictive shielding consistently reduces both hypo- and hyperglycemic events across all baselines, showing higher TIR and lower Risk Indices than both the unshielded and rule-based ones.

### G.4. Type 2 Diabetes (T2D)

Tables 12, 13, and 14 summarize the results for the T2D cohort.

**Exacerbated Failure of Reactive Heuristics.** The limitations of RBS are even worse than T1D setting. Due to the delayed glucose dynamics characteristic of T2D, reactive interventions frequently misjudge the necessary insulin suspension. For instance, applying RBS to PPO-Lag causes a huge increase in hyperglycemic events, jumping from 2.39% to 10.42%. While hypoglycemic events are reduced, the effect is a severe degradation in control: PPO-Lag's TIR falls from 80.06% to 72.57%, and Risk Index almost doubles (4.85 → 8.23). This pattern is consistent across all baselines, confirming that static rules cannot robustly handle physiological distribution shifts.

**Widening the Generalization Gap.** In contrast, Predictive Shield adapts to T2D dynamics, improving performance more than in T1D dynamics. Shielded CPO achieves a remarkable 95.22% TIR (up from 82.33%) and reduces Risk Index to a minimal 1.99. Similarly, PPO-Lag sees a massive

| Algorithm | TIR (%) | Risk Index | CV (%) | Hypo Events (%) | Hyper Events (%) |
|---|---|---|---|---|---|
| CPO | 95.22 ± 2.20 | 1.99 ± 0.53 | 19.72 ± 3.08 | 3.91 ± 1.38 | 0.42 ± 0.51 |
| CUP | 80.26 ± 6.81 | 5.36 ± 1.63 | 33.15 ± 4.62 | 4.82 ± 0.77 | 4.48 ± 2.76 |
| FOCOPS | 77.65 ± 1.20 | 5.66 ± 0.61 | 33.46 ± 1.91 | 6.36 ± 0.51 | 4.10 ± 1.45 |
| CRPO | 86.78 ± 13.03 | 4.19 ± 3.34 | 28.13 ± 11.62 | 9.33 ± 8.76 | 0.95 ± 1.04 |
| PCPO | 42.71 ± 2.16 | 18.43 ± 0.76 | 51.61 ± 1.06 | 6.48 ± 0.51 | 30.47 ± 2.15 |
| PPOLag | 93.63 ± 4.26 | 2.53 ± 1.01 | 24.10 ± 4.72 | 2.97 ± 1.35 | 1.10 ± 1.60 |
| RCPO | 84.50 ± 18.32 | 4.46 ± 4.31 | 29.51 ± 16.10 | 10.16 ± 14.07 | 1.03 ± 1.53 |
| TRPOLag | 81.71 ± 9.31 | 4.49 ± 1.99 | 27.20 ± 3.02 | 6.51 ± 1.83 | 2.99 ± 3.67 |

*Table 14.* Aggregated performance across cohorts (T2D, Predictive Shield)

| Algorithm | TIR (%) | Risk Index | CV (%) | Hypo Events (%) | Hyper Events (%) |
|---|---|---|---|---|---|
| CPO | 75.81 ± 2.82 | 6.17 ± 0.32 | 29.22 ± 1.82 | 16.47 ± 4.32 | 0.62 ± 0.74 |
| CUP | 82.03 ± 1.48 | 4.89 ± 0.45 | 29.32 ± 0.26 | 10.19 ± 0.05 | 1.17 ± 0.29 |
| FOCOPS | 75.47 ± 2.66 | 6.22 ± 0.99 | 34.03 ± 0.49 | 18.78 ± 2.05 | 1.47 ± 0.29 |
| CRPO | 74.89 ± 0.94 | 5.87 ± 0.32 | 27.88 ± 0.47 | 10.36 ± 2.35 | 1.57 ± 0.05 |
| PCPO | 43.03 ± 1.21 | 18.60 ± 0.03 | 51.30 ± 0.33 | 7.68 ± 0.60 | 30.80 ± 0.14 |
| PPOLag | 77.55 ± 3.74 | 5.97 ± 1.18 | 28.94 ± 1.28 | 17.44 ± 4.00 | 0.29 ± 0.05 |
| RCPO | 80.57 ± 0.29 | 4.98 ± 0.13 | 25.53 ± 2.63 | 12.70 ± 2.93 | 0.14 ± 0.07 |
| TRPOLag | 72.87 ± 5.07 | 7.26 ± 0.35 | 30.49 ± 5.17 | 20.79 ± 0.58 | 0.73 ± 1.22 |

*Table 15.* Aggregated performance across cohorts (T2D without Pump, No Shield)

| Algorithm | TIR (%) | Risk Index | CV (%) | Hypo Events (%) | Hyper Events (%) |
|---|---|---|---|---|---|
| CPO | 76.92 ± 4.78 | 6.64 ± 1.04 | 36.44 ± 3.46 | 3.65 ± 1.19 | 6.88 ± 1.89 |
| CUP | 78.23 ± 1.81 | 6.65 ± 0.49 | 38.32 ± 3.20 | 3.81 ± 0.68 | 7.28 ± 1.18 |
| FOCOPS | 71.31 ± 1.69 | 8.35 ± 0.48 | 47.88 ± 0.99 | 5.89 ± 0.34 | 10.96 ± 1.09 |
| CRPO | 77.18 ± 2.38 | 6.27 ± 0.45 | 33.89 ± 3.42 | 2.77 ± 0.28 | 5.72 ± 1.01 |
| PCPO | 47.26 ± 1.44 | 16.52 ± 0.51 | 52.33 ± 0.88 | 7.85 ± 0.18 | 26.36 ± 0.82 |
| PPOLag | 76.82 ± 4.10 | 7.15 ± 1.25 | 41.00 ± 3.03 | 4.19 ± 0.77 | 8.64 ± 2.50 |
| RCPO | 77.40 ± 6.08 | 6.98 ± 2.05 | 36.57 ± 6.30 | 3.73 ± 1.04 | 7.76 ± 4.26 |
| TRPOLag | 71.42 ± 2.88 | 8.65 ± 0.33 | 43.87 ± 1.16 | 4.78 ± 0.46 | 11.40 ± 1.28 |

*Table 16.* Aggregated performance across cohorts (T2D without Pump, Rule-based Shield)

| Algorithm | TIR (%) | Risk Index | CV (%) | Hypo Events (%) | Hyper Events (%) |
|---|---|---|---|---|---|
| CPO | 80.17 ± 11.31 | 5.31 ± 3.19 | 26.60 ± 10.83 | 9.69 ± 16.59 | 0.00 ± 0.00 |
| CUP | 85.00 ± 10.79 | 4.35 ± 2.42 | 25.18 ± 5.23 | 8.93 ± 12.00 | 0.74 ± 0.87 |
| FOCOPS | 80.02 ± 2.97 | 5.03 ± 0.91 | 32.63 ± 1.74 | 13.96 ± 2.18 | 1.35 ± 0.73 |
| CRPO | 86.71 ± 0.00 | 3.47 ± 0.00 | 20.35 ± 0.00 | 0.11 ± 0.00 | 0.00 ± 0.00 |
| PCPO | 45.35 ± 0.43 | 17.61 ± 0.26 | 50.93 ± 0.33 | 6.22 ± 0.20 | 29.42 ± 0.52 |
| PPOLag | 81.75 ± 5.54 | 4.94 ± 1.46 | 33.69 ± 5.76 | 9.36 ± 3.97 | 1.79 ± 1.46 |
| RCPO | 81.10 ± 6.21 | 4.98 ± 1.82 | 30.33 ± 7.24 | 9.07 ± 3.20 | 2.09 ± 3.16 |
| TRPOLag | 74.06 ± 1.52 | 6.42 ± 0.36 | 36.17 ± 0.92 | 13.95 ± 0.62 | 3.59 ± 1.30 |

*Table 17.* Aggregated performance across cohorts (T2D without Pump, Predictive Shield)

improvement, with TIR rising to 93.63% and hyperglycemic events dropping to 1.10%. Also, safety gap between the unshielded and shielded agents is wider in T2D (+13% TIR) than in T1D (+7% TIR). This validates our core hypothesis: as the test environment diverges further from the training distribution, the base policy struggles to generalize and our proposed shield improves safety metrics.

### G.5. Type 2 Diabetes without Pump

Tables 15, 16, and 17 summarize the results for the T2D without pump cohort.

**Failure of Reactive Heuristics.** Similar to the previous results, RBS effectively reduces hypoglycemia, for example, CRPO hypo events from 10.36% to 2.77%. However, it suffers from increased hyper events. Hence, the overall Risk Index for all algorithms but PCPO actually *worsens* compared to the unshielded baseline.

**Predictive Shielding Advantage.** Similar to the previous results, Predictive Shield improves TIR, Risk Index, and CV successfully. This is most evident with CRPO: TIR improves by almost 12% (reaching 86.71%), the Risk Index drops to 3.47, and severe hypoglycemic events are eliminated (0.11%).

## H. Latency and Shield Trigger Rate

We evaluate the computational overhead of the shielding mechanisms by measuring runtime under the same evaluation protocol with no shield, the predictive shield, and the rule-based shield. As shown in Table 18, the predictive shield introduces substantial latency, increasing runtime from roughly 2–11 seconds without shielding to 84–434 seconds with predictive verification. This overhead comes from repeatedly evaluating model-based safety predictions

for candidate actions.

In contrast, the rule-based shield is substantially cheaper, with runtimes between 25 and 34 seconds across algorithms. For most algorithms, this corresponds to roughly a 3× overhead relative to the unshielded policy, while being about an order of magnitude faster than the predictive shield. The exception is PCPO, whose unshielded runtime is much smaller, making its relative overhead appear larger despite a comparable absolute rule-based runtime.

The trigger rates further show that the latency difference is not explained only by how often the shield intervenes. Predictive and rule-based shields are both activated on a minority of decisions, but predictive shielding remains much slower because each intervention requires a more expensive safety prediction step. These results indicate that predictive shielding provides a stronger model-based safety check but is computationally costly, whereas the rule-based shield offers a more practical low-latency alternative.

All runtimes were measured on CPU-only servers through our SLURM-based evaluation pipeline, without GPU acceleration or inference-level optimization. These costs should also be interpreted relative to the simulator's control frequency: each environment step corresponds to a 5-minute interval. Therefore, although predictive shielding adds substantial wall-clock overhead in our current CPU implementation, this overhead may be less prohibitive in the intended low-frequency clinical decision setting than in high-frequency control domains. Still, predictive shielding is much slower than rule-based shielding, motivating future system-level optimization.

| Algorithm | No Shield Runtime (s) | Predictive Shield Runtime (s) | Rule-Based Shield Runtime (s) | Predictive Trigger (%) | Rule-Based Trigger (%) |
|---|---|---|---|---|---|
| CPO | $10.618 \pm 1.007$ | $345.816 \pm 25.241$ | $33.685 \pm 0.832$ | $28.84 \pm 2.42$ | $18.99 \pm 1.70$ |
| CUP | $10.254 \pm 1.040$ | $339.778 \pm 102.734$ | $32.291 \pm 1.292$ | $28.43 \pm 2.17$ | $21.34 \pm 2.61$ |
| FOCOPS | $10.037 \pm 0.656$ | $404.114 \pm 54.164$ | $31.972 \pm 0.244$ | $33.51 \pm 2.80$ | $25.85 \pm 0.43$ |
| OnCRPO | $9.927 \pm 1.140$ | $402.590 \pm 135.313$ | $33.291 \pm 0.654$ | $30.03 \pm 1.63$ | $19.23 \pm 0.69$ |
| PCPO | $2.062 \pm 0.193$ | $83.704 \pm 9.155$ | $25.046 \pm 0.226$ | $17.00 \pm 1.90$ | $31.04 \pm 0.41$ |
| PPOLag | $10.527 \pm 0.747$ | $434.330 \pm 22.389$ | $32.344 \pm 1.096$ | $28.38 \pm 2.21$ | $20.78 \pm 0.30$ |
| RCPO | $10.072 \pm 0.828$ | $367.563 \pm 58.554$ | $32.977 \pm 0.832$ | $28.21 \pm 4.82$ | $19.97 \pm 2.36$ |
| TRPOLag | $9.466 \pm 0.890$ | $330.079 \pm 12.249$ | $33.436 \pm 0.984$ | $33.14 \pm 3.30$ | $22.15 \pm 1.66$ |

*Table 18.* **Runtime and shield trigger rates.** Runtime and intervention frequency aggregated across diabetes types by algorithm.

# I. Stress Test under Imperfect Patient Compliance

We further stress-test the predictive shield under imperfect patient compliance. The main experiments assume full compliance, where accepted actions are followed with probability $100\%$. Here, we additionally evaluate medium compliance ($75\%$ acceptance) and low compliance ($50\%$ acceptance). For each algorithm and compliance level, we sweep the logic masking over $\{-1, -5, -10, -20\}$ in Equation 1, and report the setting with the highest mean TIR across three seeds. We omit the selected penalty from the tables to keep the comparison focused on clinical outcomes. Metrics are reported as mean $\pm$ standard deviation across three seeds, after aggregating across patients and diabetes types within each seed.

As shown in Tables 19, 20, and 21, reduced compliance makes the control problem harder: lower acceptance rates generally decrease TIR and worsen Risk Index/CV. Nevertheless, shielded policies remain meaningfully functional under partial compliance, suggesting that the predictive shield is not entirely dependent on perfect adherence.

| Algorithm | Delta TIR (%) | Shielded TIR (%) | Shielded Risk Index | Shielded (%) |
|---|---|---|---|---|
| CPO | 3.47 ± 0.20 | 83.42 ± 2.75 | 4.24 ± 0.64 | 27.31 ± 1.40 |
| CUP | 1.31 ± 1.89 | 79.34 ± 3.60 | 5.62 ± 0.94 | 33.27 ± 3.17 |
| FOCOPS | 2.49 ± 3.27 | 79.68 ± 3.43 | 5.50 ± 1.03 | 31.51 ± 1.32 |
| OnCRPO | 9.23 ± 4.22 | 86.22 ± 3.51 | 3.77 ± 0.95 | 25.30 ± 3.64 |
| PCPO | 6.20 ± 0.00 | 44.85 ± 0.00 | 17.92 ± 0.00 | 50.61 ± 0.00 |
| PPOLag | 5.64 ± 3.74 | 84.42 ± 6.55 | 4.67 ± 1.77 | 29.14 ± 5.29 |
| RCPO | 7.36 ± 7.53 | 84.27 ± 11.12 | 4.39 ± 2.85 | 27.76 ± 8.25 |
| TRPOLag | 3.49 ± 1.49 | 79.85 ± 2.97 | 5.12 ± 0.71 | 31.35 ± 1.01 |

*Table 19.* **Full compliance performance.** Aggregated predictive-shield performance across all diabetes types under 100% patient compliance. Delta TIR is predictive-shield TIR minus no-shield TIR.

| Algorithm | Delta TIR (%) | Shielded TIR (%) | Shielded Risk Index | Shielded (%) |
|---|---|---|---|---|
| CPO | 4.16 ± 0.32 | 82.65 ± 4.27 | 4.41 ± 0.94 | 28.10 ± 2.34 |
| CUP | 2.09 ± 1.21 | 78.38 ± 3.44 | 5.74 ± 0.90 | 33.65 ± 2.76 |
| FOCOPS | 3.46 ± 0.67 | 78.41 ± 0.99 | 5.46 ± 0.29 | 33.85 ± 1.11 |
| OnCRPO | 3.16 ± 0.45 | 79.24 ± 1.40 | 4.88 ± 0.37 | 29.60 ± 2.22 |
| PCPO | 1.25 ± 0.59 | 39.52 ± 0.67 | 20.18 ± 0.24 | 52.87 ± 0.44 |
| PPOLag | 5.33 ± 0.63 | 81.85 ± 1.72 | 4.60 ± 0.40 | 31.23 ± 1.88 |
| RCPO | 2.59 ± 0.67 | 78.92 ± 3.50 | 5.32 ± 1.01 | 30.50 ± 2.85 |
| TRPOLag | 3.37 ± 1.32 | 78.86 ± 2.39 | 5.29 ± 0.58 | 32.00 ± 1.38 |

*Table 20.* **Medium compliance performance.** Aggregated predictive-shield performance across all diabetes types under 75% patient compliance. Delta TIR is predictive-shield TIR minus no-shield TIR.

| Algorithm | Delta TIR (%) | Shielded TIR (%) | Shielded Risk Index | Shielded (%) |
|---|---|---|---|---|
| CPO | 3.63 ± 0.29 | 80.76 ± 3.17 | 4.75 ± 0.70 | 31.42 ± 1.30 |
| CUP | 1.91 ± 1.78 | 77.21 ± 3.03 | 6.06 ± 0.92 | 35.22 ± 2.80 |
| FOCOPS | 2.82 ± 0.78 | 78.02 ± 0.45 | 5.60 ± 0.10 | 35.16 ± 1.09 |
| OnCRPO | 2.85 ± 1.03 | 76.52 ± 1.78 | 5.52 ± 0.42 | 32.98 ± 2.94 |
| PCPO | 1.89 ± 0.43 | 39.79 ± 0.45 | 20.29 ± 0.31 | 53.40 ± 0.37 |
| PPOLag | 4.96 ± 0.58 | 80.72 ± 2.14 | 4.88 ± 0.53 | 33.10 ± 1.96 |
| RCPO | 3.53 ± 0.82 | 77.18 ± 3.16 | 5.64 ± 0.97 | 32.99 ± 3.18 |
| TRPOLag | 4.00 ± 0.62 | 77.40 ± 1.74 | 5.60 ± 0.45 | 34.02 ± 0.99 |

*Table 21.* **Low compliance performance.** Aggregated predictive-shield performance across all diabetes types under 50% patient compliance. Delta TIR is predictive-shield TIR minus no-shield TIR.

