# OpenReview forum: "Safety Generalization Under Distribution Shift in Safe Reinforcement Learning: A Diabetes Testbed"
_ICML.cc/2026/Conference — ICML 2026 regular_

### Official Review · Reviewer_qdnu · 2026-03-09

**Soundness:** 3
**Presentation:** 4
**Significance:** 4
**Originality:** 3
**Overall Recommendation:** 5
**Confidence:** 2

**Summary:**

This paper studies whether safety achieved during training in safe reinforcement learning transfers to deployment under physiological distribution shift. Using diabetes management as a safety-critical setting, the authors build a unified simulation benchmark to compare in-distribution and out-of-distribution performance, and show that policies that appear safe during training often suffer clear safety degradation on unseen patients and longer horizons. To address this, they propose a test-time predictive shielding mechanism that uses clinical rules together with a patient-specific glucose dynamics model, BA-NODE, to filter unsafe actions before execution.

**Compliance With Llm Reviewing Policy:**

Affirmed.

**Key Questions For Authors:**

see weaknesses above

**Limitations:**

yes

**Strengths And Weaknesses:**

Strengths
1. The overall framework is coherent, combining a benchmark, a patient-adaptive predictor, and a test-time safety layer in a clean way.
2. The evaluation is broad and uses clinically meaningful metrics rather than only RL reward or cost.
3. The proposed shield is practically appealing because it can be attached to existing policies without retraining.

Weaknesses
1. The benchmark may overstate the safety generalization gap. Each policy is trained on only one representative patient per cohort, so part of the observed degradation may reflect an intentionally narrow training distribution rather than a general failure of safe RL methods.
2. The main claims are supported only in simulation. The authors also acknowledge that unannounced meals or acute stress can break the forecast model, so the paper should be more careful about deployment-facing claims.
3. Its main value lies in exposing the safety generalization gap under distribution shift and in providing a practical test-time mitigation framework, rather than in introducing a fundamentally new safe RL algorithm or a strong new theory. I am not sure if this conforms to the style of ICML.

---

> ### Author Rebuttal · Authors · 2026-03-30
>
> **Effect of broader patient coverage in training.**
> We thank the reviewer for this insightful observation. We agree that training on only one representative patient creates a narrow training distribution, and this likely amplifies the observed degradation. To explain this, we compares the original setting, where the policy is trained on 1 patient and evaluated over 10 patients total (1 in-distribution + 9 out-of-distribution), against training on more patients, where the policy is trained on 4 patients and evaluated over 10 patients total (4 in-distribution + 6 out-of-distribution). Across all diabetes types, the broader-training setting shows higher TIR and lower Risk Index/CV on average. That said, we believe the original 1-patient / 9-unseen setting remains a meaningful stress test: it tests an extreme low-data setting, which is relevant in medical settings where patient data are often limited due to privacy and collection constraints.
>
> **Table** Delta metrics are more-patients results minus base-policy results, reported as mean ± std across 3 seeds after aggregating across patients within each seed.
>
> | Diabetes Type | TIR ↑ (%) | ΔTIR ↑ | Risk Index ↓ | ΔRisk ↓ | CV ↓ (%) | ΔCV ↓ |
> |---|---:|---:|---:|---:|---:|---:|
> | T1D | 87.57 ± 2.60 | 6.77 ± 4.80 | 3.31 ± 0.53 | -1.65 ± 1.24 | 25.91 ± 1.83 | -2.05 ± 1.50 |
> | T2D | 87.56 ± 0.37 | 6.04 ± 6.71 | 3.68 ± 0.15 | -1.20 ± 1.84 | 23.83 ± 0.65 | -2.82 ± 4.38 |
> | T2D w/o pump | 80.88 ± 4.01 | 6.03 ± 6.32 | 5.19 ± 0.78 | -1.18 ± 0.97 | 25.93 ± 5.87 | -3.32 ± 7.09 |
>
>
> **Simulation scope and deployment-facing claims.**
> We thank the reviewer for highlighting this. We agree and appreciate this caution. Our broader impact statement notes the risk of over-reliance on runtime shields when learned dynamics fail under unmodeled disturbances, as well as the danger of over-interpreting simulator results. However, we agree that this point should also be stated more explicitly in the main text. Our results are simulation-only, and we do not claim immediate clinical readiness. In particular, unannounced meals, acute stress, and other real-world disturbances can degrade forecast accuracy and therefore weaken shielding effectiveness. We will revise the paper to soften deployment-facing language and more clearly separate simulation evidence from claims that would require real-world validation
>
>
>
> **Our contributions.**
>  We thank the reviewer for this thoughtful comment. We agree that the paper’s primary contribution is not a fundamentally new safe RL training algorithm. Instead, its main value lies in: (1) identifying and quantifying the safety generalization gap under clinical distribution shift, (2) introducing a unified simulator and benchmark for studying this problem, and (3) proposing a practical test-time mitigation framework through predictive shielding and personalized dynamics predictor. We believe these are important machine learning contributions because they expose vulnerabilities that standard safe RL evaluation can miss, provide a reproducible benchmark for future work, and show that deployment-time safety mechanisms can improve safety under distribution shift.

---

> > ### Author Rebuttal · Reviewer_qdnu · 2026-04-03
> >
> > Thank you for resolving my concern. I believe this score is positive enough, so I will keep my score unchanged.

---

> > > ### Author Response · Authors · 2026-04-03
> > >
> > > Thank you for your thoughtful review and for considering our response.

---

### Official Review · Reviewer_Eaun · 2026-03-11

**Soundness:** 4
**Presentation:** 4
**Significance:** 4
**Originality:** 3
**Overall Recommendation:** 5
**Confidence:** 3

**Summary:**

The paper investigates the critical problem of safety generalization in Safe RL under OOD dynamics. The authors introduce a unified clinical simulator for diabetes management, covering Type 1, Type 2, and Type 2 without pump therapies, as a high-stakes testbed. Through an extensive benchmark of eight Safe RL algorithms, they empirically demonstrate a safety generalization gap, showing that policies satisfying constraints during training frequently violate them on unseen patients. To mitigate this, the authors propose a test-time predictive shielding mechanism. This shield utilizes a novel Basis-Adaptive Neural ODE (BA-NODE) to forecast glucose trajectories and proactively mask unsafe actions. The approach yields significant improvements in clinical metrics, including TIR and Risk Index, across diverse patient populations without requiring policy retraining.

**Compliance With Llm Reviewing Policy:**

Affirmed.

**Final Justification:**

The rebuttal fully solved my concerns and made the proposed method more promising. I believe my initial score of 5 can reflect the novelty and importance of the paper.

**Key Questions For Authors:**

1. Could you provide a brief empirical analysis of the inference time required for the BA-NODE predictive shield per environment step, compared to the base policy inference?

2. Could you please report the explicit number (or percentage) of actions/trajectories that were intercepted and masked by the RBS versus the Predictive Shield?

3. Could you discuss potential strategies to smooth these rescue interventions?

**Limitations:**

yes

**Strengths And Weaknesses:**

## Strengths
1. The application of Safe RL to medical control is an excellent choice for studying latent distribution shifts. The development of a unified simulator that incorporates realistic physiological variability represents an important contribution to the community.

2. The BA-NODE architecture is elegantly designed for this specific problem. By integrating variate-aware history encoding with a continuous-time Latent Neural ODE ensemble and function-space adaptation, the model inherently respects the continuous, non-linear, and patient-specific nature of glucose-insulin dynamics.

3. Testing across 72 experimental settings, three diabetes types, three age groups, and eight distinct baseline algorithms provides highly convincing evidence of both the generalization gap and the efficacy of the proposed shield.

## Weakness
1.  While the simulator models delayed action and temporal refractory periods, it assumes perfect patient compliance regarding the execution of the accepted actions. The evaluation would be stronger if the shield were stress-tested against deliberate adherence failures.

2. The proposed test-time shield must evaluate an ensemble of Neural ODEs and solve a regularized least-squares problem for function-space combination at every decision step. The manuscript lacks an analysis of the computational latency this introduces.

3. While the shield generally improves safety, the authors transparently note that in the Type 2 Diabetes without pump cohort, the shield actually worsens the CV. The paper attributes this to active rescue behavior causing sudden glucose inflections. However, a discussion on how to smooth these rescue interventions to prevent high volatility is missing.

---

> ### Author Rebuttal · Authors · 2026-03-30
>
> **Stress Test under Imperfect Patient Compliance.**
> We thank the reviewer for this suggestion. We added experiments that explicitly vary patient compliance with accepted actions. In the main paper, results assume full compliance (100%). In the rebuttal, we additionally evaluate medium compliance (75% acceptance) and low compliance (50% acceptance). Across the selected algorithms, we observe a clear pattern: lower compliance makes it harder to achieve strong clinical outcomes, leading to lower TIR and generally worse Risk Index/CV. This stress test shows that adherence failures do degrade performance, but the shielded policies still remain meaningfully functional under reduced compliance.
> Metrics are reported as mean ± std across 3 seeds, after aggregating across patients and diabetes types within each seed.
>
> Full compliance
> | Algorithm | TIR ↑ (%) | Risk Index ↓ | CV ↓ (%) |
> |-----------|----------------|----------------|----------------|
> | CPO | 83.42 ± 2.75 | 4.24 ± 0.64 | 27.31 ± 1.40 |
> | CUP | 79.34 ± 3.60 | 5.62 ± 0.94 | 33.27 ± 3.17 |
> | OnCRPO | 86.22 ± 3.51 | 3.77 ± 0.95 | 25.30 ± 3.64 |
>
> Medium compliance
> | Algorithm | TIR ↑ (%) | Risk Index ↓ | CV ↓ (%) |
> |-----------|----------------|----------------|----------------|
> | CPO | 82.65 ± 4.27 | 4.41 ± 0.94 | 28.10 ± 2.34 |
> | CUP | 78.38 ± 3.44 | 5.74 ± 0.90 | 33.65 ± 2.76 |
> | OnCRPO | 79.24 ± 1.40 | 4.88 ± 0.37 | 29.60 ± 2.22 |
>
> Low compliance
> | Algorithm | TIR ↑ (%) | Risk Index ↓ | CV ↓ (%) |
> |-----------|----------------|----------------|----------------|
> | CPO | 80.76 ± 3.17 | 4.75 ± 0.70 | 31.42 ± 1.30 |
> | CUP | 77.21 ± 3.03 | 6.06 ± 0.92 | 35.22 ± 2.80 |
> | OnCRPO | 76.52 ± 1.78 | 5.52 ± 0.42 | 32.98 ± 2.94 |
>
> **Computational Latency, Per-Step Inference, and Shield Trigger Rates.**
> We thank the reviewers for raising this point. We added an empirical runtime and trigger-rate analysis for the unshielded controller, RBS, and Predictive Shield with a 24-step horizon prediction (120 minutes ahead). Experiments were run on a CPU-only server over a full 7-day episode (2016 steps). The table reports total episode runtime, per-step inference time (episode runtime ÷ 2016 steps, in ms), and shield trigger rate (percentage of actions intercepted).
>
> | Algorithm | Base Runtime (s) | Base Per-Step (ms) | Predictive Shield Runtime (s) | Predictive Per-Step (ms) | RBS Runtime (s) | RBS Per-Step (ms) | Predictive Trigger (%) | RBS Trigger (%) |
> |-------------|------------------|--------------------|-------------------------------|--------------------------|-----------------|-------------------|------------------------|-----------------|
> | CPO | 10.61 ± 1.00 | 5.26 ± 0.50 | 345.81 ± 25.24 | 171.53 ± 12.52 | 33.68 ± 0.83 | 16.71 ± 0.41 | 28.84 ± 2.42 | 18.99 ± 1.70 |
> | CUP | 10.25 ± 1.04 | 5.08 ± 0.52 | 339.77 ± 102.73 | 168.54 ± 50.96 | 32.29 ± 1.29 | 16.02 ± 0.64 | 28.43 ± 2.17 | 21.34 ± 2.61 |
> | OnCRPO | 9.92 ± 1.14 | 4.92 ± 0.57 | 402.59 ± 135.31 | 199.70 ± 67.12 | 33.29 ± 0.65 | 16.51 ± 0.32 | 30.03 ± 1.63 | 19.23 ± 0.69 |
>
> Predictive shielding increases runtime from ~10 s to 340–400 s per episode (30–40× overhead) in this CPU execution, while RBS requires ~32–34 s. The Predictive Shield intercepts 28–30% of actions, compared to 19–21% for RBS. We view this as an important limitation of the current long-horizon CPU execution. However, this can be mitigated by high performant GPU execution, batching, shorter horizons, and smaller models for predictions.
>
> **Discussion of the inconsistent CV results and smoothing rescue interventions.**
>  We thank the reviewer for highlighting this. Our interpretation is that the inconsistent CV behavior reflects an asymmetry in personalization between the predictive shield and the base RL policy. The shield uses BA-NODE, which adapts predictions using patient history and context, whereas the underlying Safe RL policy is only state-conditioned and does not explicitly perform patient-specific adaptation. As a result, under distribution shift, the base policy may still propose actions that are suboptimal for the current patient even when the shield successfully prevents unsafe outcomes. Since the shield is designed to filter actions predicted to violate safety thresholds, it aligns most directly with improvements in TIR and Risk Index, while CV measures glucose variability and is not directly optimized by the shielding rule. Thus, shielding can reduce severe failures while having a less uniform effect on variability.
> To smooth rescue behavior, several extensions are possible: adding penalties on abrupt action changes or frequent interventions, replacing hard rescue decisions with softer risk-aware action reweighting, incorporating trajectory-smoothness objectives into the rescue rule, or combining shielding with a more adaptive history-conditioned base policy. We will clarify this discussion in the revision.

---

> > ### Author Rebuttal · Reviewer_Eaun · 2026-04-01
> >
> > Thank you very much for the detailed rebuttal. I am fully convinced by your explanations regarding the computational latency and the inconsistent CV behavior (Answers 2 and 3). Regarding the runtime analysis, I would actually argue that the time cost of your current CPU-based execution is already acceptable for real-world deployment. Given that even continuous glucose monitors take measurements on a 1 to 5-minute interval, a per-step predictive inference latency of ~170–200 ms falls well within practical clinical constraints and shouldn't be viewed as a major limitation from the application perspective.
> >
> > However, I require some clarification regarding the patient compliance stress test in Answer 1. It is not clear to me if the tables provided represent the performance of the shield-based versions or the base algorithms. Are the currently reported results with the BA-NODE? If so, how do the original, unshielded RL algorithms perform under these exact same reduced compliance conditions (75% and 50%)? To fully understand the robustness provided by your method in these scenarios, could you please clarify this and explicitly report the changes in TIR and Risk Index comparing the algorithms with and without the BA-NODE shield?

---

> > > ### Author Response · Authors · 2026-04-02
> > >
> > > **Follow-up on Patient compliance stress test / comparison with and without shielding using BA-NODE.** We thank the reviewer for the helpful follow-up. The compliance-stress-test tables previously reported in the rebuttal correspond to the *shielded* setting, i.e., the base RL controller augmented with the predictive shield using BA-NODE. To directly isolate the contribution of the shield under reduced compliance, we now additionally report the corresponding *no-shield* results under the same compliance settings, together with the paired change from no-shield to shielded performance. As requested, we focus on *TIR* and *Risk Index*, and report all results as mean ± std across 3 seeds.
> > >
> > > ### Full compliance
> > >
> > > | Algorithm | No-shield TIR (%) | No-shield Risk Index | Shielded TIR (%) | Shielded Risk Index | ΔTIR (%) | ΔRisk Index |
> > > | --------- | ----------------: | -------------------: | ---------------: | ------------------: | -------: | ----------: |
> > > | CPO       | 79.94 ± 2.56 | 5.21 ± 0.66 | 83.42 ± 2.75 | 4.24 ± 0.64 | 3.47 ± 0.20 | 0.96 ± 0.18 |
> > > | CUP       | 78.04 ± 5.50 | 5.85 ± 1.48 | 79.34 ± 3.60 | 5.62 ± 0.94 | 1.31 ± 1.89 | 0.23 ± 0.56 |
> > > | OnCRPO    | 76.99 ± 2.83 | 5.65 ± 0.59 | 86.22 ± 3.51 | 3.77 ± 0.95 | 9.23 ± 4.22 | 1.88 ± 1.00 |
> > >
> > > ### Medium compliance
> > >
> > > | Algorithm | No-shield TIR (%) | No-shield Risk Index | Shielded TIR (%) | Shielded Risk Index | ΔTIR (%) | ΔRisk Index |
> > > | --------- | ----------------: | -------------------: | ---------------: | ------------------: | -------: | ----------: |
> > > | CPO       | 78.48 ± 4.49 | 5.50 ± 1.12 | 82.65 ± 4.27 | 4.41 ± 0.94 | 4.16 ± 0.32 | 1.09 ± 0.35 |
> > > | CUP       | 76.29 ± 4.58 | 6.21 ± 1.25 | 78.38 ± 3.44 | 5.74 ± 0.90 | 2.09 ± 1.21 | 0.47 ± 0.42 |
> > > | OnCRPO    | 76.08 ± 1.85 | 5.78 ± 0.42 | 79.24 ± 1.40 | 4.88 ± 0.37 | 3.16 ± 0.45 | 0.90 ± 0.30 |
> > >
> > > ### Low compliance
> > >
> > > | Algorithm | No-shield TIR (%) | No-shield Risk Index | Shielded TIR (%) | Shielded Risk Index | ΔTIR (%) | ΔRisk Index |
> > > | --------- | ----------------: | -------------------: | ---------------: | ------------------: | -------: | ----------: |
> > > | CPO       | 77.13 ± 3.11 | 5.87 ± 0.94 | 80.76 ± 3.17 | 4.75 ± 0.70 | 3.63 ± 0.29 | 1.12 ± 0.35 |
> > > | CUP       | 75.30 ± 4.68 | 6.50 ± 1.46 | 77.21 ± 3.03 | 6.06 ± 0.92 | 1.91 ± 1.78 | 0.44 ± 0.56 |
> > > | OnCRPO    | 73.67 ± 2.80 | 6.44 ± 0.72 | 76.52 ± 1.78 | 5.52 ± 0.42 | 2.85 ± 1.03 | 0.93 ± 0.45 |
> > >
> > > Overall, shielding with BA-NODE consistently improves TIR and reduces Risk Index across all reported algorithms and compliance settings. We will incorporate this clarification and the corresponding with/without-shield comparison into the revision.

---

### Official Review · Reviewer_gLW9 · 2026-03-13

**Soundness:** 3
**Presentation:** 3
**Significance:** 3
**Originality:** 2
**Overall Recommendation:** 5
**Confidence:** 3

**Summary:**

This paper examines whether safety during training generalizes to test time under distributional shifts in the context of diabetes management. To that end, the authors develop a clinical simulator for patients with type 1 and 2 diabetes that accounts for patient variability. An evaluation of safe RL algorithms reveals a consistent safety generalization gap, which the paper addresses via a test-time predictive shielding mechanism that masks actions predicted to cause safety violations, using a learned dynamics model called BA-NODE. Across many settings, the predictive shielding improves time-in-range while reducing clinical risk.

**Compliance With Llm Reviewing Policy:**

Affirmed.

**Final Justification:**

I maintain my positive assessment of the paper.

**Key Questions For Authors:**

- Have the authors looked into the sensitivity of the shielding mechanism to the complexity of the learned dynamics model? If so, what did they find? If not, how would they do it?
- Is there a benefit to training on more patients, rather than one, as the former sounds more realistic?
- Since the main focus is on safety gap under distribution shift, why didn't the authors look into robust approaches?

**Limitations:**

Yes.

**Strengths And Weaknesses:**

Strengths

- The authors design rigorous experiments across 72 settings, where training is on a single patient and testing is on nine unseen patients, showcasing the generalization gap. The results evidence that training-time safety does not generalize, and TIR drops in all baselines.
The paper is well-structured and has a logical flow. Research questions in the experiments section clearly highlight what to look for in the results.

- Safety generalization gap is an important problem that is not only constrained to diabetes management, but this particular setting comes with a natural distribution shift. The empirical findings raise important questions for the safe RL literature. The simulator and benchmark provide a setting to work for future research.

- Although there is no algorithmic novelty in this work, the main originality lies in the problem formulation, scientific findings, and the benchmark.

Weaknesses

- The paper doesn’t provide any ablations into the learned dynamics model, e.g., on its complexity.

- The inconsistency in the CV results should be discussed more.

- Although the authors provide many safe RL baselines, a better comparison would be against methods designed for robustness under distribution shift, as that is the main problem they focus on.

---

> ### Author Rebuttal · Authors · 2026-03-30
>
> **Sensitivity to dynamics model complexity.**
> Yes. Shielding depends on prediction quality, consistent with Theorem 5.2’s $(\epsilon,\alpha)$-reliability view: improved prediction accuracy corresponds to a smaller error bound $\epsilon$, which supports more reliable predictive filtering.
>
> In the paper, Fig. 11 in Appendix shows robustness to basis-number choice. In rebuttal, we added a hidden-size sweep $\{16,32,64,128\}$ (hidden size 128 is used for our main experiments). The results are reported in Table below. Prediction quality is fairly stable across hidden sizes, with modest improvement as hidden size increases. Together, these results suggest that BA-NODE is robust to moderate changes in model capacity.
>
> **Table**: Sensitivity of BA-NODE to hidden size on dynamics prediction
>
> | Hidden size | MAE ↓ | FDE ↓ | RMSE ↓ |
> |-------------|--------------|--------------|--------------|
> | 16 | 2.87 ± 0.02 | 3.72 ± 0.04 | 4.56 ± 0.08 |
> | 32 | 2.87 ± 0.06 | 3.69 ± 0.09 | 4.58 ± 0.23 |
> | 64 | 2.85 ± 0.02 | 3.69 ± 0.04 | 4.49 ± 0.04 |
> | 128 | **2.82 ± 0.01** | **3.63 ± 0.03** | **4.42 ± 0.02** |
>
> **Impact of Training on More Patients.**
> Yes. As expected, training on more patients improves performance. The table below compares the original setting, where the policy is trained on 1 patient and evaluated over 10 patients total (1 in-distribution + 9 out-of-distribution), against training on more patients, where the policy is trained on 4 patients and evaluated over 10 patients total (4 in-distribution + 6 out-of-distribution). Across all diabetes types, the broader-training setting shows higher TIR and lower Risk Index/CV on average. That said, we believe the original 1-patient / 9-unseen setting remains a meaningful stress test: it tests an extreme low-data setting, which is relevant in medical settings where patient data are often limited due to privacy and collection constraints.
>
> **Table** Delta metrics are more-patients results minus base-policy results, reported as mean ± std across 3 seeds after aggregating across patients within each seed.
>
> | Diabetes Type | TIR ↑ (%) | ΔTIR ↑ | Risk Index ↓ | ΔRisk ↓ | CV ↓ (%) | ΔCV ↓ |
> |---|---:|---:|---:|---:|---:|---:|
> | T1D | 87.57 ± 2.60 | 6.77 ± 4.80 | 3.31 ± 0.53 | -1.65 ± 1.24 | 25.91 ± 1.83 | -2.05 ± 1.50 |
> | T2D | 87.56 ± 0.37 | 6.04 ± 6.71 | 3.68 ± 0.15 | -1.20 ± 1.84 | 23.83 ± 0.65 | -2.82 ± 4.38 |
> | T2D w/o pump | 80.88 ± 4.01 | 6.03 ± 6.32 | 5.19 ± 0.78 | -1.18 ± 0.97 | 25.93 ± 5.87 | -3.32 ± 7.09 |
>
> **Robust Safe RL Approaches and Complementarity of Shielding.**
> We thank the reviewer for this important point. Robust Safe RL approaches are indeed highly relevant for handling distribution shift. Our goal in this paper, however, was first to identify and quantify the safety generalization gap of standard Safe RL methods under latent physiological shift, rather than to provide an exhaustive comparison against all robustness-oriented training approaches. In that sense, the unified clinical simulator is intended not only to reveal this gap, but also to serve as a benchmark for evaluating future robust methods in the safety critical domain.
> Our predictive shield is complementary to such methods: it is an algorithm-agnostic runtime wrapper that requires no retraining and can be layered on top of standard Safe RL, robust Safe RL, or other controllers. We therefore expect robust training and shielding to be synergistic: robust training can improve the base policy, while shielding provides an extra deployment-time safeguard under distribution shift. Evaluating robust Safe RL, alone and combined with shielding, is an important future direction.
>
>
> **Discussion of the inconsistent CV results.**
> We thank the reviewer for highlighting this.
> Our interpretation is that this reflects an asymmetry in personalization between the predictive shield and the base RL policy. The shield uses BA-NODE, which adapts predictions using patient history and context. However, the underlying Safe RL policy is a state-conditioned controller trained on a limited patient distribution and does not explicitly perform patient-specific adaptation. As a result, under distribution shift, the base policy may still propose actions that are suboptimal for the current patient even when the shield successfully prevents the unsafe outcomes.
> This distinction matters because the shield is designed to filter actions that are predicted to violate safety thresholds, which aligns most directly with improvements in TIR and Risk Index. In contrast, CV measures glucose variability, which is not optimized by the shielding rule. Thus, while shielding can reduce severe failures, its effect on variability can be less uniform and may depend on the quality and adaptability of the underlying controller.
> We will revise the discussion to make this point clearer and to note that combining predictive shielding with a more adaptive base policy may lead to more consistent improvements in CV.

---

> > ### Author Rebuttal · Reviewer_gLW9 · 2026-04-03
> >
> > I thank the authors for the rebuttal.
> >
> > Dynamics model sensitivity:
> >
> > The hidden-size sweep and the existing basis-number ablation demonstrate that BA-NODE is robust to changes in model complexity. Prediction quality remains stable across a range of hidden sizes, with modest improvements as capacity increases. This is consistent with the theoretical framing where better prediction corresponds to a tighter error bound in the shielding guarantee.
> >
> > Training on more patients:
> >
> > The experiment comparing 1-patient and 4-patient training confirms the expected benefits of broader training data, with improvements in TIR and Risk Index across all diabetes types. I agree that the original 1-patient setting remains a meaningful stress test for low-data medical scenarios.
> >
> > Robust Safe RL approaches:
> >
> > The clarification that the paper's goal is to identify and quantify the safety generalization gap, rather than to benchmark robust methods, is reasonable. Positioning the predictive shield as a complementary, algorithm-agnostic runtime wrapper that can be layered on top of robust approaches is convincing, and acknowledging this combination as future work is satisfactory.
> >
> > CV inconsistency:
> >
> > The explanation linking inconsistent CV results to the asymmetry between the adaptive shield and the non-adaptive base policy is clear. The shield targets safety threshold violations, which aligns with TIR and Risk Index improvements, while CV captures variability that the shielding rule does not directly optimize. I appreciate the commitment to revising the discussion accordingly.
> >
> > I maintain my positive assessment of the paper.

---

> > > ### Author Response · Authors · 2026-04-04
> > >
> > > Thank you for your thoughtful review and for considering our response.

---

### Decision · Program_Chairs · 2026-04-30

**Decision:**

Accept (regular)

**Comment:**

The paper addresses an important problem of generalization in Safe RL under OOD dynamics. The work is overall technically solid, clearly written and the authors have provided substantial clarifications and further validation in the rebuttal phase to address the majority of the reviewer concerns. My recommendation is thus to accept the paper.